# ImageReward: Learning and Evaluating Human Preferences for Text-to-Image Generation

**Jiazheng Xu**[◇†*], **Xiao Liu**[◇‡*], **Yuchen Wu**[◇], **Yuxuan Tong**[◇], **Qinkai Li**[§†], **Ming Ding**[‡],
**Jie Tang**[◇], **Yuxiao Dong**[◇]

[◇]Tsinghua University     [‡]Zhipu AI     [§]Beijing U. of Posts and Telecommunications

## Abstract

We present a comprehensive solution to learn and improve text-to-image models from human preference feedback. To begin with, we build ImageReward—the first general-purpose text-to-image human preference reward model—to effectively encode human preferences. Its training is based on our systematic annotation pipeline including rating and ranking, which collects 137k expert comparisons to date. In human evaluation, ImageReward outperforms existing scoring models and metrics, making it a promising automatic metric for evaluating text-to-image synthesis. On top of it, we propose Reward Feedback Learning (ReFL), a direct tuning algorithm to optimize diffusion models against a scorer. Both automatic and human evaluation support ReFL's advantages over compared methods. All code and datasets are provided at `https://github.com/THUDM/ImageReward`.

## 1 Introduction

Text-to-image generative models, including auto-regressive [43; 11; 14; 16; 12; 63] and diffusion-based [37; 45; 42; 46] approaches, have experienced rapid advancements in recent years. Given appropriate text descriptions (i.e., prompts), these models can generate high-fidelity and semantically-related images on a wide range of topics, attracting significant public interest in their potential applications and impacts.

Despite the progress, existing self-supervised pre-trained [33] generators are far from perfect. A primary challenge lies in **aligning models with human preference**, as the pre-training distribution is noisy and differs from the actual user-prompt distributions. The inherent discrepancy leads to several well-documented issues in the generated images [15; 31], including but not limited to:

- **Text-image Alignment**: failing to accurately depict all the numbers, attributes, properties, and relationships of objects described in text prompts, as shown in Figure 1 (a)(b).
- **Body Problem**: presenting distorted, incomplete, duplicated, or abnormal body parts (e.g., limbs) of humans or animals, as illustrated in Figure 1 (e)(f).
- **Human Aesthetic**: deviating from the average or mainstream human preference for aesthetic styles, as demonstrated in Figure 1 (c)(d).
- **Toxicity and Biases**: featuring content that is harmful, violent, sexual, discriminative, illegal, or causing psychological discomfort, as depicted in Figure 1 (f).

These prevalent challenges, however, are difficult to address solely through improvements in model architectures and pre-training data.

In natural language processing (NLP), researchers have employed reinforcement learning from human feedback (RLHF) [55; 36; 39] to guide large language models [6; 7; 66; 48; 64] towards human preferences and values. The approach relies on learning a reward model (RM) to capture human

---

[*]JX & XL contributed equally. Corresponding authors: YD & JT (`yuxiaod|jietang@tsinghua.edu.cn`).
[†]Work done when JX interned at Zhipu AI and QL visited Tsinghua University.

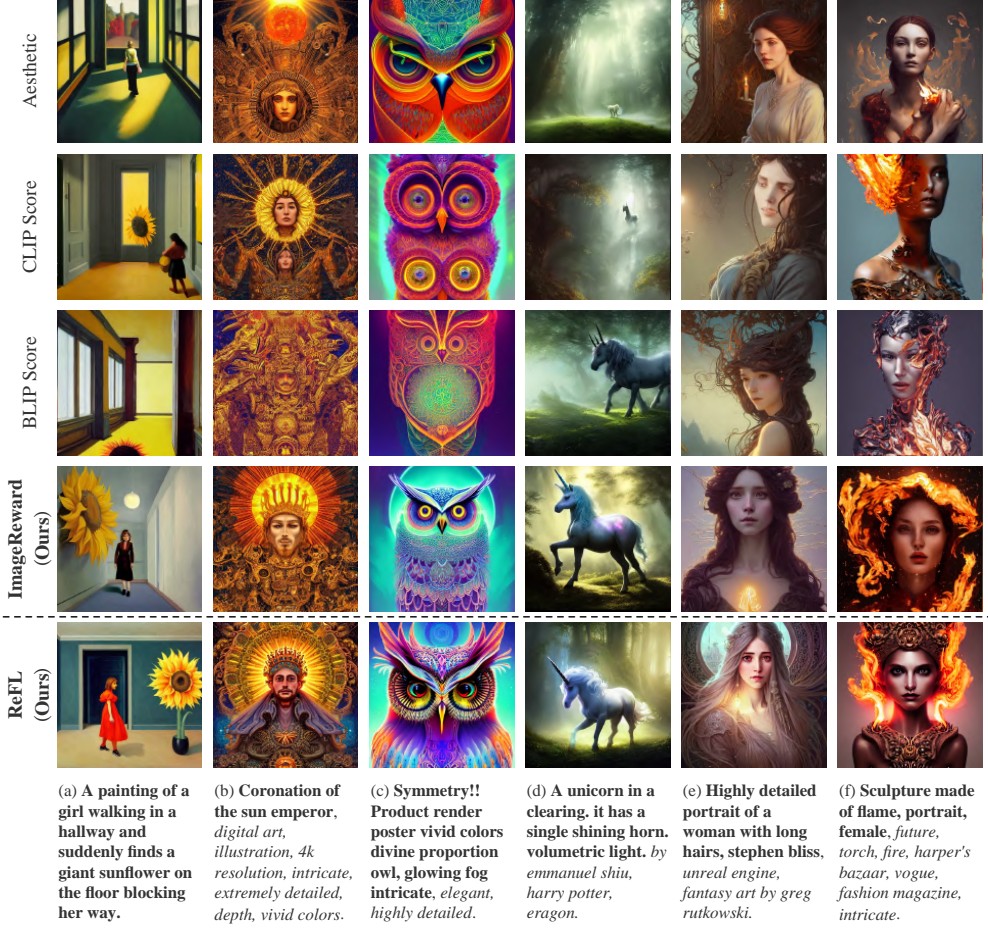

Figure 1: (Upper) Top-1 images out of 64 generations selected by different text-image scorers. (Lower) 1-shot generation after ReFL training using ImageReward as feedback. Images get better text coherence and human preference after ImageReward selection or ReFL training. In prompts (from real users, truncated), the **bold** roughly denotes content, and the *italic* denotes style or function.

preference from massive expert-annotated model output comparisons. Effective though it is, the annotation process can be costly and challenging [39], as it requires months of effort to establish labeling criteria, recruit and train experts, verify responses, and ultimately produce the RM.

**Contributions.** Recognizing the importance of addressing these challenges in generative models, we present and release the first general-purpose text-to-image human preference RM—ImageReward— which is trained and evaluated on 137k pairs of expert comparisons in total, based on real-world user prompts and corresponding model outputs. Based on the effort, we further investigate the direct optimization approach ReFL for improving diffusion generative models. Our main contributions are:

- We systematically identify the challenges for text-to-image human preference annotation, and consequently design a pipeline tailored for it, establishing criteria for quantitative assessment and annotator training, optimizing labeling experience, and ensuring quality validation. We build the text-to-image comparison dataset for training the ImageReward model based on the pipeline. The overall architecture is depicted in Figure 2.

- We demonstrate that ImageReward outperforms existing text-image scoring methods, such as CLIP [41] (by 38.6%), Aesthetic [50] (by 39.6%), and BLIP [26] (by 31.6%), in terms of understanding human preference in text-to-image synthesis through extensive analysis and experiments. ImageReward is also proven to significantly mitigate the aforementioned issues, providing valuable insights into how human preference can be integrated into generative models.

- We suggest that ImageReward could serve as a promising automatic text-to-image evaluation metric. Compared to FID [18] and CLIP scores on prompts from real users and MS-COCO 2014, ImageReward aligns consistently to human preference ranking and presents higher distinguishability across models and samples.

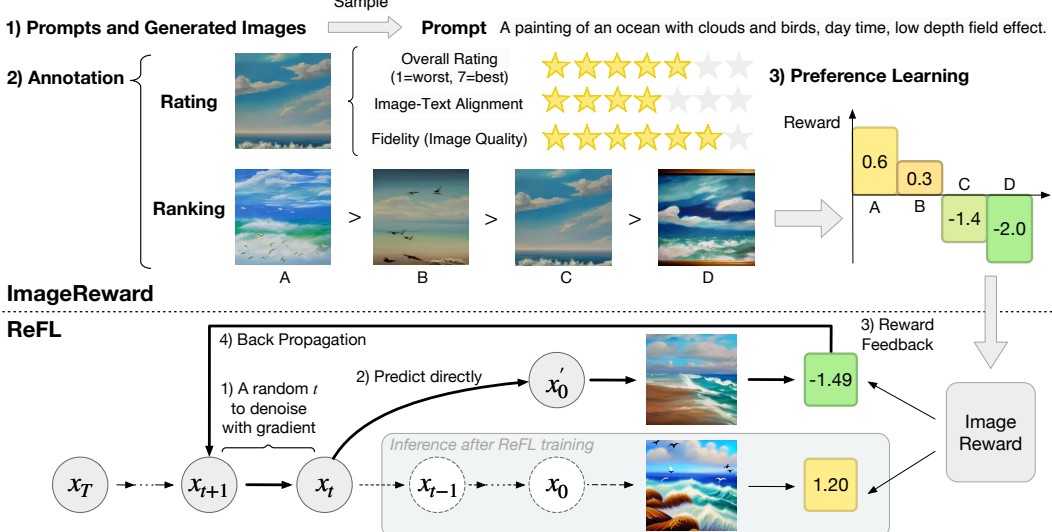

Figure 2: An overview of the ImageReward and ReFL. (Upper) ImageReward's annotation and training, consisting of data collection, annotation, and preference learning. (Lower) ReFL leverages ImageReward's feedback to directly optimize diffusion models at a random latter denoising step.

- We propose Reward Feedback Learning (ReFL) for tuning diffusion models regarding human preference scorers. Our unique insight on ImageReward's quality identifiability at latter denoising steps allows the direct feedback learning on diffusion models, which offer no likelihood for their generations. Extensive automatic and human evaluations demonstrate ReFL's advantages over existing approaches including data augmentation [61; 13] and loss reweighing [23].

## 2 ImageReward: Learning to Score and Evaluate Human Preferences

ImageReward is constructed using a systematic pipeline involving data collection and human annotation from experts. Based on the pipeline, we implement the RM training and derive the ImageReward.

### 2.1 Annotation Pipeline Design

**Prompt Selection and Image Collection.** The dataset utilizes a diverse selection of real user prompts from DiffusionDB [58], an open-sourced dataset. To ensure diversity in selected prompts, we employ a graph-based algorithm that leverages language model-based prompt similarity [56; 44; 53]. This selection yields 10,000 candidate prompts, each accompanied by 4 to 9 sampled images from DiffusionDB, resulting in 177,304 candidate pairs for labeling (Cf. Appendix A.1 for details).

**Human Annotation Design.** Our annotation pipeline involves a prompt annotation stage, which includes categorizing prompts and identifying problematic ones, and a text-image rating stage, where images are rated based on *alignment*, *fidelity*, and *harmlessness*. Subsequently, annotators rank the images in order of preference. To manage potential contradictions in the ranking, we provide trade-offs in our annotation document (completely attached in Appendix B). Our annotation system is composed of three stages: Prompt Annotation, Text-Image Rating, and Image Ranking. Screenshots of our system are provided in Figure 8. Annotators were recruited in collaboration with a professional data annotation company, with a majority having at least college-level education. They are trained using documents that describe the labeling process and criteria. To ensure quality, we employ quality inspectors to double-check each annotation, with invalid annotations reassigned for relabeling. Due to the page limits, please refer to Appendix A.3, A.2, B for comprehensive details and discussion.

**Human Annotation Analysis.** After 2 months of annotation, we collected valid annotations for 8,878 prompts, resulting in 136,892 compared pairs. A comprehensive analysis of these prompts, annotations, and challenges discovered is discussed in detail in Appendix A.4.

Table 1: Text-to-image model ranking by humans and automatic metrics (ImageReward, CLIP, and FID). *Zero-shot FID (30k) scores of DALL-E 2 is taken from [42]; others are evaluated in 256×256 resolution on MS-COCO 2014 validation set following prior practices.

| Dataset & Model | Real User Prompts | | | | | | MS-COCO 2014 | | | |
| | Human Eval. | | ImageReward | | CLIP | | ImageReward | | Zero-shot FID* | |
| | Rank | #Win | Rank | Score | Rank | Score | Rank | Score | Rank | Score |
| --- | --- | --- | --- | --- | --- | --- | --- | --- | --- | --- |
| Openjourney | 1 | 507 | 1 | 0.2614 | 2 | 0.2726 | 3 | -0.0455 | 5 | 20.7 |
| Stable Diffusion 2.1-base | 2 | 463 | 2 | 0.2458 | 4 | 0.2683 | 2 | 0.1553 | 4 | 18.8 |
| DALL-E 2 | 3 | 390 | 3 | 0.2114 | 3 | 0.2684 | 1 | 0.5387 | 1 | 10.9* |
| Stable Diffusion 1.4 | 4 | 362 | 4 | 0.1344 | 1 | 0.2763 | 4 | -0.0857 | 2 | 17.9 |
| Versatile Diffusion | 5 | 340 | 5 | -0.2470 | 5 | 0.2606 | 5 | -0.5485 | 3 | 18.4 |
| CogView 2 | 6 | 74 | 6 | -1.2376 | 6 | 0.2044 | 6 | -0.8510 | 6 | 26.2 |
| Spearman $\rho$ to Human Eval. | | - | | 1.00 | | 0.60 | | 0.77 | | 0.09 |

## 2.2 RM Training

Admittedly, human evaluation is after all the touchstone for human preference for synthesized images; but it is limited by labor costs and hard to scale up. We aim to model human preference based on annotations, which can lead to a virtual evaluator free from dependence on humans.

Similar to RM training for language model of previous works [55; 39], we formulate the preference annotations as rankings. We have $k \in [4, 9]$ images ranked for the same prompt $T$ (the best to the worst are denoted as $x_1, x_2, ..., x_k$) and get at most $C_k^2$ comparison pairs if no ties between two images. For each comparison, if $x_i$ is better and $x_j$ is worse, the loss function can be formulated as:

$$\text{loss}(\theta) = -\mathbb{E}_{(T,x_i,x_j)\sim\mathcal{D}}[\log(\sigma(f_\theta(T, x_i) - f_\theta(T, x_j)))] \tag{1}$$

where $f_\theta(T, x)$ is a scalar value of preference model for prompt $T$ and generated image $x$.

**Training Techniques.** We use BLIP [26] as the backbone of ImageReward, as it outperforms conventional CLIP (Cf. Table 2b) in our preliminary experiments. We extract image and text features, combine them with cross attention, and use an MLP to generate a scalar for preference comparison.

Training ImageReward is of no ease. We observe rapid convergence and consequent overfitting, which harms its performance. To address this, we freeze some backbone transformer layers' parameters, finding that a proper number of fixed layers improves ImageReward's performance (Cf. Section 4.1). ImageReward also exhibits sensitivity to training hyperparameters, such as learning rate and batch size. We perform a careful grid search based on the validation set to determine optimal values.

## 2.3 As Metric: Re-Evaluating Human Preferences on Text-to-Image Models

Training text-to-image generative models is hard, but evaluating these models reasonably is even harder. In literature [11; 42; 12; 46], it has been a *de facto* practice to evaluate text-to-image generative models on MS-COCO [28] image-caption dataset against the real images, using fine-tuned or zero-shot FID [18] scores following DALL-E [43] setting. Nevertheless, it remains quite dubious whether the FID really fits the current need [38], especially from the following aspects:

1. **Zero-shot Usage**: As generative models are now dominantly used by the public in a zero-shot manner without fine-tuning, fine-tuned FID may not honestly reflect models' actual performance in real use. In addition, despite the adoption of zero-shot FID in recent trends, the possible leak of MS-COCO in some models' pre-training data would make it a potentially unfair setting.
2. **Human Preference**: FID measures the average distance between generated images and reference real images, and thus fails to encompass human preference that is crucial to text-to-image synthesis in evaluation. Moreover, FID's relies on average over the whole dataset to provide an accurate assessment, whereas in many cases we need the metric to serve as a selector over single images.

Seeing these challenges, we propose ImageReward as a promising zero-shot automatic evaluation metric for text-to-image model comparison and individual sample selection.

**Better Human Alignment Across Models.** We conduct researcher annotation (i.e., by authors) across 6 popular high-resolution (around 512×512) available text-to-image models: CogView 2 [12],

Versatile Diffusion (VD) [62], Stable Diffusion (SD) 1.4 and 2.1-base [45], DALL-E 2 (via OpenAI API) [42], and Openjourney[1], to identify the alignment of different metrics to human.

We sample 100 real-user test prompts for the alignment test, with each model generating 10 outputs as candidates. To compare these models, we first pick the best image out of 10 outputs by each model on each prompt. Then, the annotators rank the images from different models for each prompt, following the disciplines for ranking described in Section 2.1. We aggregate all annotators' annotations, and compute the final win count of each model to all others (Cf. Table 1).

For ImageReward and CLIP scores, we report their average for 1,000 text-image pairs per model. We also document all models' zero-shot FID and ImageReward score (30k) on MS-COCO 2014 valid set following prior practices [42; 12], where outputs are unified to 256×256 resolution and optimal classifier-free guidance values are selected by grid search (i.e., [1.5, 2.0, 3.0, 4.0, 5.0]). As shown in Table 1, ImageReward aligns well with human ranking, whereas zero-shot FID and CLIP are not.

**Better Distinguishability Across Models and Samples.** Another highlight is that, compared to CLIP, we observe that ImageReward can better distinguish the quality between individual samples. Figure 3 presents a box plot of ImageReward and CLIP's score distributions on the 1,000 generations per model. The distributions are normalized to 0.0 to 1.0 using minimum and maximum values of ImageReward and CLIP scores per model, and outliers are discarded. As it demonstrates, ImageReward's scores in each model have a much larger interquartile range than that of CLIP, which means ImageReward can well distinguish the quality of images from each other. Besides, in terms of comparison across models, we discover that the medians of the ImageReward scores are also roughly in line with human ranking in Table 1. On the contrary, CLIP's medians fail to present the property.

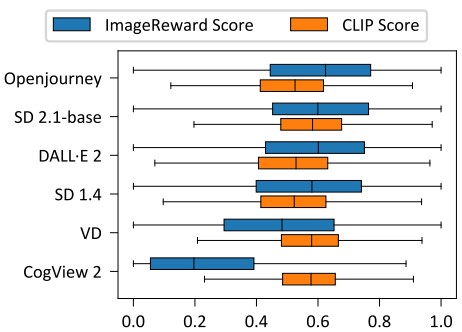

Figure 3: Normalized distribution of ImageReward and CLIP scores of different generative models (outliers are discarded). ImageReward's scores align well with human preference and present higher distinguishability.

## 3 ReFL: Reward Feedback Learning Improves Text-to-Image Diffusion

Though ImageReward can pick out highly human-preferred images from many generations of a prompt, the generate-and-then-filter paradigm could be expensive and inefficient in practical applications. Therefore, we seek to improve text-to-image generative models, particularly for the popular latent diffusion models, for allowing high-quality generation in single or very few trials.

**Challenge.** In NLP, researchers have reported using reinforcement learning algorithms (e.g., PPO [51]) to steer language models to align to human preference [55; 36; 39], which depends on the likelihood of a whole generation to update the model.

However, unlike language models, latent diffusion models (LDMs)'s multi-step denoising generation cannot yield likelihoods for their generations, and thus fail to adopt the same RLHF approaches. A potentially similar approach is classifier-guidance [54; 9] technique during LDM inference. Nonetheless, it is for inference only

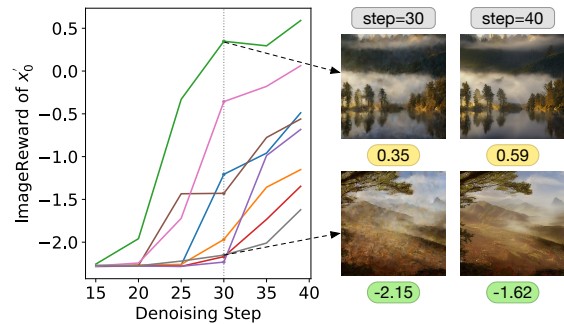

Figure 4: ImageReward scores of a prompt with different generation seeds along denoising steps. Final image qualities become identifiable after 30 out of 40 steps.

---

[1] https://openjourney.art/

and employs a classifier necessarily trained on noisy intermediate latents, which naturally contradicts RMs' annotation where images need to be completely denoised for humans to mark correct preference. Some concurrent works propose some alternative indirect solutions, such as using RMs to filter dataset for fine-tuning [61; 13], or to re-weight losses of training samples according to their qualities [23]. Nevertheless, these data-oriented approaches are virtually indirect. They could rely heavily on proper fine-tuning data distributions and finally only improve the LDMs mildly.

**ReFL: Insight and Solution.** We endeavor to develop a direct optimization method for improving LDMs according to an RM (e.g., ImageReward). Looking into ImageReward scores along denoising steps (i.e., 40 in our case), we derive an intriguing insight (Cf. Figure 4) that when we directly predict $x_t \to x_0'$ at a step $t$ (different from the real latent $x_0$ which experiences $x_t \to x_{t+1} \to ... \to x_0$):

- When $t \leq 15$: ImageReward scores for all generations are unanimously low.
- When $15 \leq t \leq 30$: High-quality generations begin to stand out, but overall we still cannot clearly judge all generations' final qualities based on the current ImageReward scores.
- When $t \geq 30$: Generations of different ImageReward scores are generally distinguishable.

---

**Algorithm 1** Reward Feedback Learning (ReFL) for LDMs

1: **Dataset:** Prompt set $\mathcal{Y} = \{y_1, y_2, ..., y_n\}$
2: **Pre-training Dataset:** Text-image pairs dataset $\mathcal{D} = \{(\text{txt}_1, \text{img}_1), ...(\text{txt}_n, \text{img}_n)\}$
3: **Input:** LDM with pre-trained parameters $w_0$, reward model $r$, reward-to-loss map function $\phi$, LDM pre-training loss function $\psi$, reward re-weight scale $\lambda$
4: **Initialization:** The number of noise scheduler time steps $T$, and time step range for fine-tuning $[T_1, T_2]$
5: **for** $y_i \in \mathcal{Y}$ and $(\text{txt}_i, \text{img}_i) \in \mathcal{D}$ **do**
6: $\quad \mathcal{L}_{pre} \leftarrow \psi_{w_i}(\text{txt}_i, \text{img}_i)$
7: $\quad w_i \leftarrow w_i$ // Update LDM$_{w_i}$ using Pre-training Loss
8: $\quad t \leftarrow rand(T_1, T_2)$ // Pick a random time step $t \in [T_1, T_2]$
9: $\quad x_T \sim \mathcal{N}(0, I)$ // Sample noise as latent
10: $\quad$ **for** $j = T, ..., t+1$ **do**
11: $\quad\quad$ **no grad:** $x_{j-1} \leftarrow$ LDM$_{w_i}\{x_j\}$
12: $\quad$ **end for**
13: $\quad$ **with grad:** $x_{t-1} \leftarrow$ LDM$_{w_i}\{x_t\}$
14: $\quad x_0 \leftarrow x_{t-1}$ // Predict the original latent by noise scheduler
15: $\quad z_i \leftarrow x_0$ // From latent to image
16: $\quad \mathcal{L}_{reward} \leftarrow \lambda\phi(r(y_i, z_i))$ // ReFL loss
17: $\quad w_{i+1} \leftarrow w_i$ // Update LDM$_{w_i}$ using ReFL loss
18: **end for**

---

In light of the observation, we conclude that ImageReward scores for generations $x_0'$ after 30 steps of denoising, unnecessarily the final step, could serve as reliable feedback for improving LDMs.

We thus propose an algorithm to directly fine-tune LDMs by viewing the scores of an RM as human preference losses to back-propagate gradients (Cf. Algorithm 1) to a randomly-picked latter step $t$ (in our case $t \in [30, 40]$) in the denoising process. The reason for the random selection of $t$ instead of using the last step is that, if only the gradient of the last denoising step is retained, the training is proved very unstable and the results are bad. In practice, to avoid rapid overfitting and stabilize the fine-tuning, we re-weight ReFL loss and regularize with pre-training loss. The final loss form is written as

$$\mathcal{L}_{reward} = \lambda \mathbb{E}_{y_i \sim \mathcal{Y}}(\phi(r(y_i, g_\theta(y_i)))) \tag{2}$$

$$\mathcal{L}_{pre} = \mathbb{E}_{(y_i, x_i) \sim \mathcal{D}}(\mathbb{E}_{\mathcal{E}(x_i), y_i, \epsilon \sim \mathcal{N}(0,1), t}[\|\epsilon - \epsilon_\theta(z_t, t, \tau_\theta(y_i))\|_2^2]) \tag{3}$$

where $\theta$ denotes the parameters of the LDM, $g_\theta(y_i)$ denotes the generated image of LDM with parameters $\theta$ corresponding to prompt $y_i$. Meanings of other symbols are detailed in Algorithm 1, while the loss function of $\mathcal{L}_{pre}$ is taken from [45].

## 4 Experiment

### 4.1 ImageReward: On Human Preference Prediction

**Dataset & Training Setting.** Rankings of annotated images are collected to train ImageReward, which contains 8,878 prompts and 136,892 pairs of image comparisons. We divide the dataset according to prompts annotated by different annotators and select 466 prompts from annotators who have a higher agreement with researchers to consist for the model test. Except for prompts for testing, other more than 8k prompts of annotation are collected for training.

Table 2: Data annotation agreement and ablation study on model backbones and dataset sizes.

(a) Agreement between different annotators, researchers, and models. Especially, "annotator ensemble" means, for each pair of images, we use the image considered better by most people as the better one.

|  | researcher | annotator | annotator ensemble | CLIP Score | Aesthetic | BLIP Score | Ours |
|---|---|---|---|---|---|---|---|
| researcher | 71.2% ± 11.1% | 65.3% ± 8.5% | 73.4% ± 6.2% | 57.8% ± 3.6% | 55.6% ± 3.1% | 57.0% ± 3.0% | **64.5%** ± **2.5%** |
| annotator | 65.3% ± 8.5% | 65.3% ± 5.6% | 53.9% ± 5.8% | 54.3% ± 3.2% | 55.9% ± 3.1% | 57.4% ± 2.7% | **65.3%** ± **3.7%** |
| annotator ensemble | 73.4% ± 6.2% | 53.9% ± 5.8% | - | 54.4% ± 21.1% | 57.5% ± 15.9% | 62.0% ± 16.1% | **70.5%** ± **18.6%** |

(b) Ablation study for different backbones (i.e., CLIP and BLIP) used in ImageReward.

| Backbone | Training Set Size | Preference Acc. |
|---|---|---|
| CLIP | 4k | 61.87 |
|  | 8k | 62.98 |
| BLIP | 1k | 63.07 |
|  | 2k | 63.18 |
|  | 4k | 64.71 |
|  | 8k | **65.14** |

Table 3: Results of ImageReward and comparison methods on human preference prediction. Preference accuracy is from the test set of 466 prompts (6,399 comparisons); Recall and Filter's scores are from another test set of 371 prompts with 8 images each. All scores are averaged per prompt.

| Model | Preference Acc. | Recall | | | Filter | | |
|---|---|---|---|---|---|---|---|
|  |  | @1 | @2 | @4 | @1 | @2 | @4 |
| CLIP Score | 54.82 | 27.22 | 48.52 | 78.17 | 29.65 | 51.75 | 76.82 |
| Aesthetic Score | 57.35 | 30.73 | 53.91 | 75.74 | 32.08 | 54.45 | 76.55 |
| BLIP Score | 57.76 | 30.73 | 50.67 | 77.63 | 33.42 | 56.33 | 80.59 |
| ImageReward (Ours) | **65.14** | **39.62** | **63.07** | **90.84** | **49.06** | **70.89** | **88.95** |

We load the pre-trained checkpoint of BLIP (ViT-L for image encoder, 12-layers transformer for text encoder) as the backbone of ImageReward, and initialize MLP head according to $\mathcal{N}(0, 1/(d_{model}+1))$ decaying the learning rate with a cosine schedule. We sweep over several value settings of learning rate and batch size and fix different rates of backbone transformer layers. We find that fixing 70% of transformer layers with a learning rate of 1e-5 and batch size of 64 can reach up to the best preference accuracy. ImageReward is trained on 4 40GB NVIDIA A100 GPUs, with a per-GPU batch size of 16.

We use the CLIP score, Aesthetic score, and BLIP score as baselines to compare with the ImageReward. CLIP score and BLIP score are calculated directly as cosine similarity between text and image embedding, while the Aesthetic score is given by an aesthetic predictor introduced by LAION[50].

**Agreement Analysis.** Agreement assesses the likelihood of two individuals sharing consistent preferences for superior images. While most people generally agree on image quality, variations in model-generated images may lead to divergent judgments. Before assessing model performance, it's crucial to measure the likelihood of consensus in selecting superior images. We use other 40 prompts (778 pairs) to calculate preference agreement between researchers, annotators, annotator ensemble, and models. Table 2a shows the result.

**Main Results: Preference Accuracy.** Preference accuracy is the correctness of a scorer choosing the same one from two different images of one prompt with a human. As Table 3 shows, our model outperforms all the baselines. The preference accuracy of ImageReward reaches up to 65.14%, which is 15.14% more than 50% (random), about twice as much as 7.76% (that of BLIP score).

**Main Results: Human Evaluation.** To evaluate the ability of ImageReward to select the more preferred images among large amounts of generated images, we produce another dataset, collecting prompts with 9/25/64 generated images from DiffusionDB, and use different methods to select from those images to get top3 results. Then three annotators rank these selected top-3 images. Figure 5 shows the win rates. Qualitative results can be seen in Appendix G, showing that ImageReward can select images that are more aligned to text and with higher fidelity and avoid toxic contents.

**Ablation Study: Training dataset size.** To investigate the effect of training dataset sizes on the performance of the model, comparative experiments are conducted. Table 2b shows that adding up the scale of the dataset significantly improves the preference accuracy of ImageReward. It's promising that if we collect more annotation data in the future, ImageReward will get better performance.

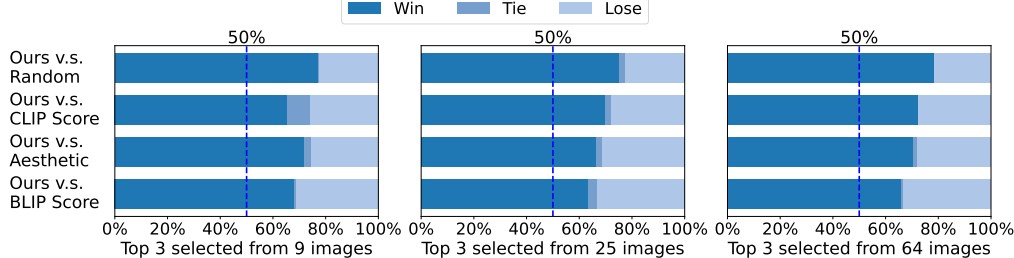

Figure 5: Win rates of ImageReward compared to other models. ImageReward wins most of the time. On average, 77.1% to random, 69.3% to CLIP, 69.8% to Aesthetic, and 65.8% to BLIP.

| Methods | Real User Prompts | | MT Bench [40] | |
|---|---|---|---|---|
| | #Win | WinRate | #Win | WinRate |
| SD v1.4 (baseline) [45] | 1315 | - | 718 | - |
| Dataset Filtering [61] | 1394 | 55.17 | 735 | 51.72 |
| Reward Weighted [23] | 1075 | 39.52 | 585 | 43.33 |
| RAFT [13] (iter=1) | 1341 | 49.86 | 578 | 42.31 |
| RAFT (iter=2) | 753 | 30.85 | 452 | 33.02 |
| RAFT (iter=3) | 398 | 20.97 | 355 | 26.19 |
| **ReFL (Ours)** | **1508** | **58.79** | **808** | **58.49** |

Table 4: Human evaluation on different LDM optimization methods. ReFL performs the best with regard to total win count and WinRate against SD v1.4 baseline.

Figure 6: Win rates between all methods.

**Ablation Study: RM backbone.** ImageReward adopts BLIP as the backbone, which may raise curiosity about how well BLIP compares to CLIP. We add MLP to CLIP, training in a similar way, and the result is also shown in Table 2b. Even if CLIP uses a relatively larger training data set, its preference is still inferior to that of BLIP. The difference between these two as backbone may partly be because BLIP used bootstrapping of its training set. Moreover, we use BLIP's image-grounded text encoder as a feature encoder different from the separate encoder for text/image as CLIP.

## 4.2    ReFL: On Improving Diffusion Models with Human Preference

**Training Settings.** We use Stable Diffusion v1.4[45] as the baseline generative model and fine-tune it for experiments. For the dataset, the pre-training dataset is from a 625k subset of LAION-5B[50] selected by aesthetic score, while the prompt set for ReFL is sampled from DiffusionDB. The model is fine-tuned in half-precision on 8 40GB NVIDIA A100 GPUs, with a learning rate of 1e-5 and batch size of 128 in total (64 for pre-training and 64 for ReFL). For ReFL algorithm, we set $\phi = ReLU, \lambda = 1e - 3$ and $T = 40, [T_1, T_2] = [1, 10]$.

**Evaluation Settings.** We collect 466 real user prompts from DiffusionDB and 90 designed challenging prompts from multi-task benchmark (MT Bench) [40] for evaluation. All fine-tuning methods use the same dataset as the pre-training dataset or generated dataset (both contain 20,000 samples), and train for one epoch with the same training settings (the same learning rate and batch size) for a fair comparison. The human evaluation is consistent with Section 2.3 and the form of dataset labeling, which involves humans sorting multiple images under a prompt. Table 4 and Figure 6 show the comparison results. All methods use the same pre-trained Stable Diffusion v1.4 and the same reward model ImageReward, using PNDM [30] noise scheduler and default classifier free guidance scale of 7.5 for inference.

We compare several important related methods for improving text-to-image generation [61; 23; 13], whose implementation details are provided in Appendix E. Compared to ReFL's direct tuning, these previous methods are all based on indirect data augmentation or loss reweighing.

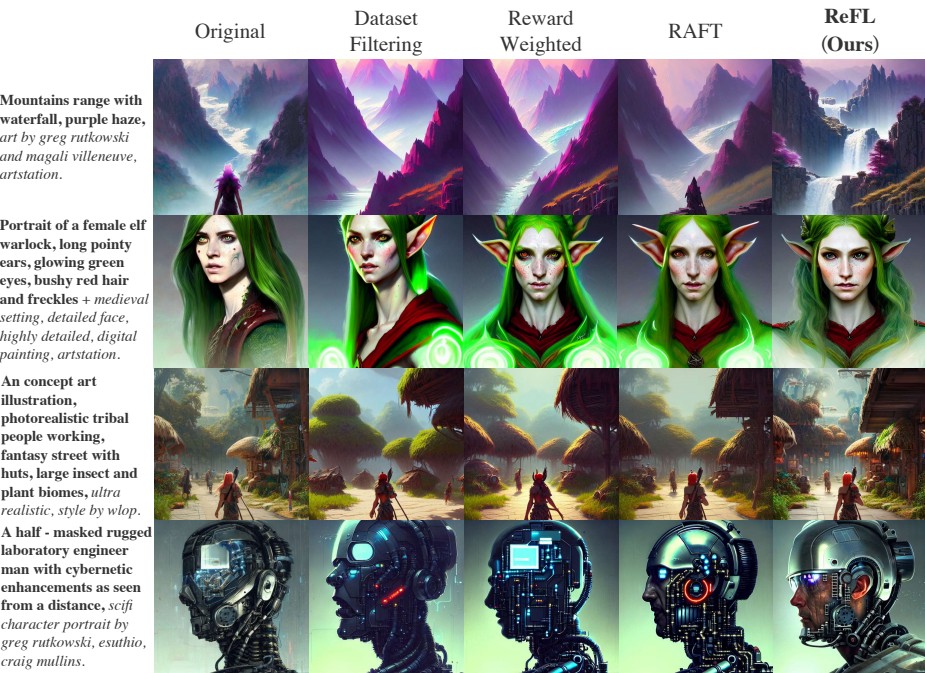

|  | Original | Dataset Filtering | Reward Weighted | RAFT | **ReFL (Ours)** |

**Mountains range with waterfall, purple haze,** *art by greg rutkowski and magali villeneuve, artstation.*

**Portrait of a female elf warlock, long pointy ears, glowing green eyes, bushy red hair and freckles** + *medieval setting, detailed face, highly detailed, digital painting, artstation.*

**An concept art illustration, photorealistic tribal people working, fantasy street with huts, large insect and plant biomes,** *ultra realistic, style by wlop.*

**A half - masked rugged laboratory engineer man with cybernetic enhancements as seen from a distance,** *scifi character portrait by greg rutkowski, esuthio, craig mullins.*

Figure 7: Qualitative comparison between ReFL and other fine-tuning methods. ReFL fine-tuned model can produce images that are more preferred overall. For example, in the second row of prompts containing "long pointy ears," only the model fine-tuned with ReFL generates correct ears, while other images either lack ears or have inaccurate representations.

**Results and analysis.** When compared to the original version, ReFL fine-tuned model is mostly preferred with the most win rate and the highest win rate. When compared to each other, ReFL is always the preferred one.

Note that in our evaluation, neither RAFT [13] nor Reward Weighted [23] has become better compared to the baseline, although they have been verified in their own experiments. Note that both RAFT and Reward Weighted do not collect the prompts used by users in real scenarios at finetune, whereas Reward Weighted manually constructs a dataset to address the alignment issue by combining colors, numbers, backgrounds, and objects. The prompts used in our review are more widely distributed and complex, so the problems with their methods are more clearly exposed.

RAFT [13] suffers from over-fitting as the number of iterations increases. [13] propose using an expert generator as a regularizer to avoid overfitting the reward model. However, RAFT is constrained by the quality of the constructed dataset. It is important to note that even expert generators have limitations, and when fine-tuning is performed using prompts sampled from real user data, which can be challenging, there may be instances where the expert generator fails to generate high-quality images.

In the case of the Reward Weighted method [23], although real images are used for regularization, there is a problem with the coefficient used for the rewards, which is constrained within the [0, 1] range. This implies that while preferred images are given larger weights and poor images are given smaller weights, the influence of the non-preferred images is not completely eliminated. Similarly, when utilizing real user prompts, it is likely that there will be non-preferred images (even those relatively the best) in the dataset, which can introduce interference. The failure to eliminate the impact of non-preferred images hinders the effectiveness of the Reward Weighted method.

Dataset Filtering [61], on the other hand, considers real images and handles non-preferred images by labeling them as "Weird image." However, this influence is indirect. In contrast, our proposed algorithm provides direct gradient feedback through rewards, allowing for guidance toward a "better" direction, which enables more effective problem-solving.

In summary, RAFT is constrained by the limited ability of the generator, and the Reward Weighted method suffers from the influence of non-preferred images due to the choice of reward coefficients.

Dataset Filtering partially addresses the problem by considering real images and labeling abnormal images, but it is still indirect and limited. By directly incorporating rewards into the gradient feedback, our proposed algorithm ReFL offers a more effective solution to these challenges. Qualitative examples are in Figure 7.

## 5 Related Work

**Text-to-image Generation and Evaluation.** Text-to-image generation has come a long way since the popularization of GANs [17], with key developments including models like DALL-E [43] and CogView [11]. Recently, diffusion models [52; 19; 10; 47] have achieved remarkable results, with Stable Diffusion [45] being particularly popular. Evaluation metrics such as Inception Score (IS) [4] and Fréchet Inception Distance (FID) [18] are commonly used to assess model performance after fine-tuning, but they cannot evaluate either single image generations or text-image coherence.

For evaluating individual generated images based on a prompt, prior works [42; 46; 63] often use CLIP [41] to calculate text-image similarity. While these metrics are useful, they don't capture human preference comprehensively. Other predictors, like Aesthetic from LAION [50], partially contribute to this holistic evaluation by scoring image aesthetics using a CLIP-based architecture. RM in RLHF, on the other hand, considers a mixture of elements such as text-image alignment, fidelity, and aesthetics. Overall, RM such as ImageReward provides a more complete evaluation for individual text-to-image generations, making it better aligned with human preferences.

**Learning from Human Feedback.** There is often a gap between generative models' pre-training objectives and human intent. Thus human feedback has been utilized to align model performance with intent in various language applications [1; 22; 36; 67; 65] via training an RM [35; 8; 59; 20; 24] to learn human preference. Researchers have explored RL for language models to achieve more truthful, helpful, and harmless outcomes [39; 2; 68; 60; 55; 29; 49; 3]. Previous work [5; 55] used human feedback to train reward functions for summarization tasks, while InstructGPT [39] applied RLHF to GPT-3 for multi-task NLP, yielding significant improvements.

In text-to-image generation, however, there have been few studies on the topic. One concurrent work [23] has focused on text-image coherence in the closed domain using simple synthetic prompts based on templates, and propose to improve models using loss re-weighing. Other concurrent works [61; 21; 13] collects 1-of-n selection from noisy online user clicking, and thus do not enforce consistent standards and prompt diversity. Their optimization methods are based on indirect data filtering and augmentation. On the contrary, ImageReward serves as a general-purpose human preference scorer with quality ensured by rigorous annotation pipeline, and corresponding ReFL is the first direct tuning method for optimize diffusion models from scorer feedback.

## 6 Conclusion

In this work, we have presented ImageReward and ReFL, the first general-purpose text-to-image human preference reward model, and a direct fine-tuning approach for optimizing diffusion models by ImageReward feedback. Through our systematic pipeline for human preference annotation, we curate a dataset of 137k expert comparisons to train ImageReward, and build ReFL algorithm on top of it. They together address prevalent issues in generative models and help to better align text-to-image generation with human values and preferences.

## Acknowledgement

We would like to thank the data annotators for their help and support. This research was supported by the Technology and Innovation Major Project of the Ministry of Science and Technology of China under Grant 2022ZD0118600 and 2022ZD0118601, Natural Science Foundation of China (NSFC) for Distinguished Young Scholars No. 61825602, NSFC No. 62276148, a research fund from Zhipu.AI, and the New Cornerstone Science Foundation through the XPLORER PRIZE.

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

# Part I

# Appendix

## Table of Contents

# A Details on ImageReward's Comparison Data Annotation Pipeline

## A.1 Prompt Selection

Training human preference RM requires a diverse prompt distribution that could cover and represent users' authentic usage. DiffusionDB has 1.8M prompts which are far beyond the number we plan to annotate. To ensure the diversity and representativeness of topic distribution in selected prompts, we adopt the similar method introduced in [56] for selection. During the graph-based selection, every sample, which is prompt in our case, is represented by a vector calculated by Sentence-BERT [44]. Then a graph is constructed with represented samples as vertices and every vertex is connected to $k$ nearest neighbors, where $k$ is a hyper-parameter in the algorithm and $k = 150$ is found to perform well. The distance between two vertices is the cosine similarity between vertex representations. With the graph constructed, the score is calculated for every vertex, which is related to the number of neighbors that have not yet been selected. Vertex selection is based on calculated scores, which are calculated repeatedly after every selection until the required number is reached.

Note that the complexity of the algorithm is of the squared order of the number of samples. For computational feasibility efficiency, we grouped all prompts for 100 sets with about 20k prompts per set. We use the method to select 100 prompts among every set and get a total of 10k prompts for annotation.

## A.2 Annotation Management

We cooperate with a professional annotation company to complete professional annotation. Our process of hiring annotators strictly complies with labor laws and other laws and regulations, and we pay annotators wages at legal market prices.

Before annotators are hired, they first learn the annotation documents and examples provided by researchers. Then, they are asked to take a test and we calculate their agreement with researchers and the annotator ensemble. Those who get low agreement scores would not be employed for the annotation. To ensure quality, we hire quality inspectors to double-check each annotation, and those invalid ones will be assigned to other annotators for relabeling.

In the final list of annotating experts, 95.8% of experts have finished at least college-level education. Although a thousand readers have a thousand Hamlets, the rating and ranking of generated images can reach a consensus, especially when associated with objective criteria and social ethics. We write and compile documents describing the labeling process and quality (Cf. Appendix B), which serve as the standard for training annotators. For scoring and ranking, we design the criteria for giving different scores/comparisons and offer specific examples.

## A.3 Human Annotation Design

Although an individual can easily identify his or her preference for a pair of images, a group of people can hardly reach consensual criteria over a massive number of comparisons in a pragmatic annotation. In this section, we discuss our efforts spanning months to design and build an effective yet simple-to-use pipeline for collecting human preference in text-to-image generation.

**Prompt Annotation.** Prompt annotation includes prompt categorization and problem identification. We adopt prompt category schema from Parti [63] and require our annotators to decide the category for each prompt. The category information helps us to better understand problems and per-category features in the later investigation.

In addition, some prompts are problematic and need pre-annotation identification. For example, some are identified as ambiguous and unclear (e.g., "a brand new medium", "low quality", etc). Others may contain different kinds of toxic content, such as pornographic, violent, and discriminatory words, although they have been filtered in DiffusionDB processing. Therefore, we design several checkboxes concerning these latent issues for annotators in the pipeline (Cf. Appendix B).

**Text-Image Rating.** Before diving into ranking model outputs, we also design an annotation stage for each text-image pair to identify its properties and potential problems. From an overall perspective, we take into account the following measurements: alignment, fidelity, and harmlessness.

- **Alignment**: which requires generated images faithfully show accurate objects of accurate attributes, with relationships between objects and events described in prompts being correct.
- **Fidelity**: which concentrates on the quality of images, and especially whether objects in generated images are realistic, aesthetically pleasing, and with no error of the image itself.
- **Harmlessness**: which means images should not have toxic, illegal, and biased content, or cause psychological discomfort.

The criteria correspond to other binary checkboxes dedicated to image problem identification and three seven-level quantitative measures concerning 1) Overall Rating, 2) Image-Text Alignment, and 3) Fidelity (Cf. Figure 8 (a)).

**Image Ranking.** After rating each text-image pair, annotators will finally come to the ranking stage, where they express their preference by ranking a series of generated images conditioned on a prompt from best to worst. The ranking generally follows the criteria mentioned in Image Rating.

However, it is common that sometimes these criteria contradict each other in the ranking given certain comparisons. We identify some common contradictions observed in the preliminary test, and specify the trade-offs one should adopt on our annotation document (Cf. Appendix B). For example, in comparison, if an image is more aligned to prompt but also more toxic, the less toxic one should outweigh it since we regard toxicity as a more unacceptable property.

**Annotation System Design.** Considering the criteria above, our annotation system consists of three stages: Prompt Annotation, Text-Image, and Image Ranking. The screenshots of our system are shown in Figure 8. The procedures for annotators to go through a prompt are as follows:

1. Label the checkboxes and enter the category for the text prompt.
2. Annotate each image one by one. Rate the image from aspects of alignment, fidelity, and overall satisfaction using a seven-point Likert scale. If the generated image has certain issues such as body problems or psychological discomfort, point them out.
3. Rank all images generated from the same prompt. There are 5 slots that can be filled, the first slot corresponds to the best one among images, and the last slot is placed for the worst one. Ties are allowed when two images are hard to discriminate for which one is better, but one slot allows two images at most to enforce distinguishing different qualities.

### A.4 Human Annotation Analysis

Among 10k prompts selected for annotation, after the expert annotation mentioned in Section 2.1, we finally collected 8,878 pieces of valid prompts, which comprise a total of 136,892 compared pairs.

**Prompt categories distribution.** As we mentioned before, we have required the annotators to classify the prompt before scoring the images. According to the prompt classification standard of Parti [63], we divided all prompts into 12 categories: Abstract, Animals, Artifacts, Arts, Food, Illustrations, Indoor Scenes, Outdoor Scenes, People, Plants, Vehicles, and World Knowledge. The distribution of the prompts in our annotation data is shown in Figure 9. As we can see, the distribution is diverse yet representative. Most prompts fall into common topics such as People

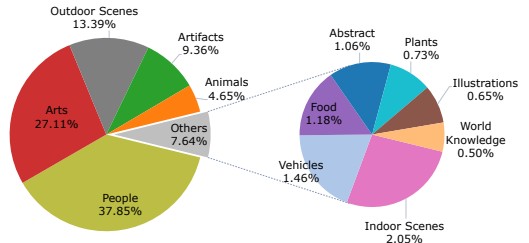

Figure 9: Prompt distribution in annotation data, which comprises 12 categories and 8,878 prompts.

(3,360), Arts (2,407), Outdoor Scenes (1,189), Artifacts (831), and Animals (413). Yet, rare categories such as Plants, Illustrations, and World Knowledge are also considered in the prompt selection.

**Average score distribution of different prompt categories.** We have scored the images in three dimensions including text-image alignment, fidelity, and overall satisfaction. The average scores of each category are shown in Figure 10. Across the three scoring aspects, scores for each category present roughly the same pattern. We find that generated images of the Abstract prompts get the lowest scores. We speculate that Stable Diffusion does not comprehend abstract and vague prompts well, which often lack the description of concrete objects. Besides, we notice that more low-quality

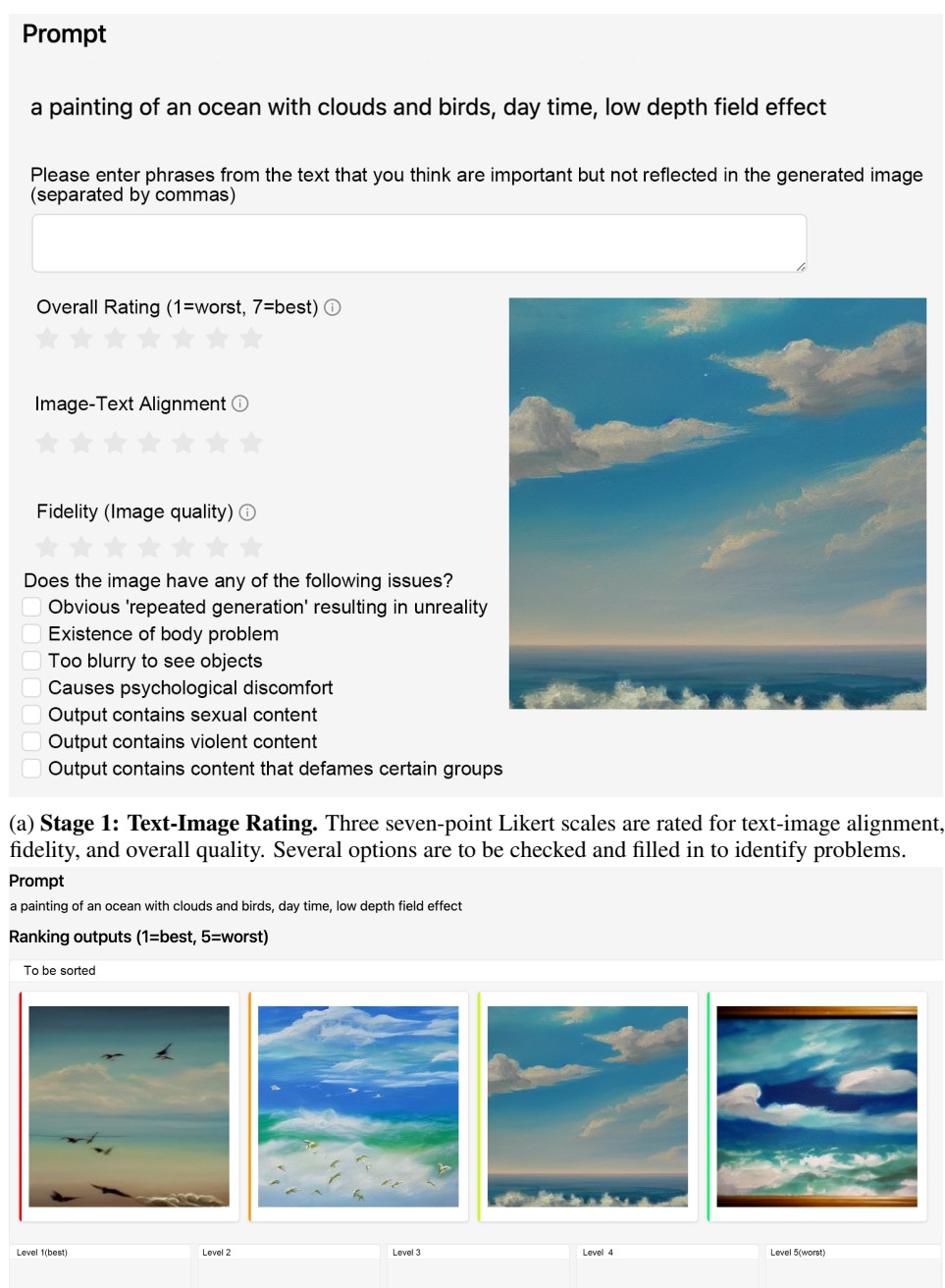

(a) **Stage 1: Text-Image Rating.** Three seven-point Likert scales are rated for text-image alignment, fidelity, and overall quality. Several options are to be checked and filled in to identify problems.

(b) **Stage 2: Image Ranking.** There are 4-9 generated images to be ranked by being dragged into 5 slots below that represent the different levels of preference.

Figure 8: Screenshots of our annotation system. Annotators first annotate each text-image pair and then finish ranking all images conditioned on the text prompt.

prompts exist in the Abstract category than in others, which may also affect the performance of the text-to-image generation. Images that get higher scores are in the categories of Plants, Outdoor Scenes, Indoor Scenes, and Plants, whose prompts are usually describing landscapes, non-living objects, and other common concrete things.

**Problem distribution of different prompt categories.** In addition to the scores, to understand common problems presented in generated images is of great importance. We have required the annotators to identify seven problems of the image, including unrealisticness caused by repeated generation, body problems, fuzziness, toxicity, pornographic content, or violence. We report the

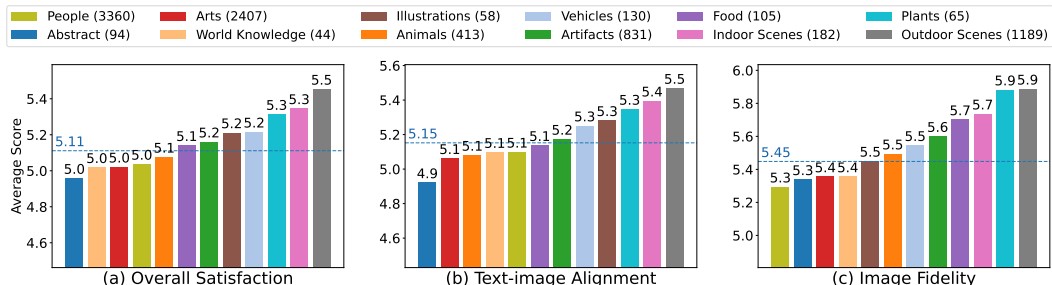

Figure 10: Average scores of each category. Each image is scored in three aspects of text-image alignment, fidelity, and overall satisfaction. For each category, we calculated the average scores of corresponding images in these three aspects. Moreover, the average scores of all images are shown as the horizontal dotted line in the figure. The number in the legend shows the number of prompts in each category.

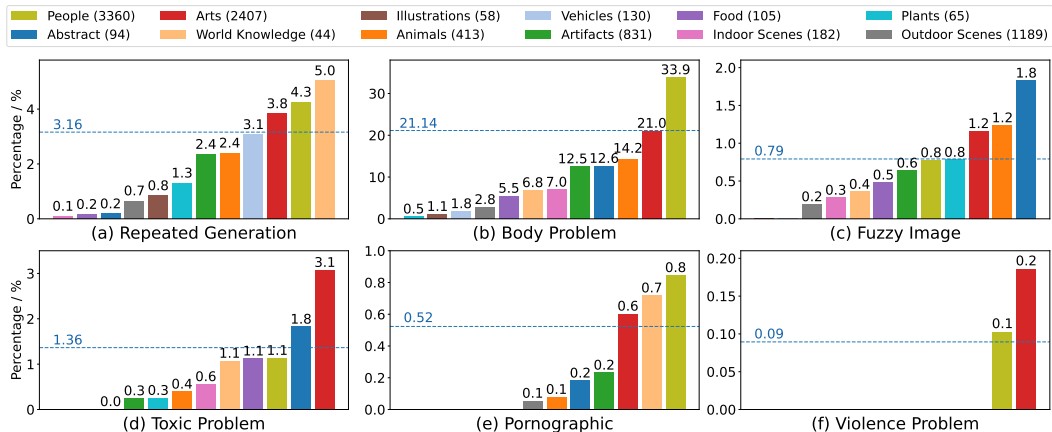

Figure 11: Problem frequency in each category. For each image, annotators are required to identify whether there are problems including unrealistic caused by repeated generation, body problems, fuzzy images, toxic, pornographic, violent, or violating protected groups. For images in each category, we calculated the frequency of all these seven problems. Especially, no images are found violating protected groups, so we omit this figure. Moreover, the frequency of these problems among all images is shown as the horizontal dotted line.

frequency of each problem in each category in Figure 11. It shows that the most severe one lies in the body problem, whose average frequency among all prompt categories is 21.14%. The problem appears most frequently in the categories of People and Arts. The Animals, Abstract, and Artifacts categories take second place. Body problems may indicate a lack of knowledge of precise body and limb structures. And this may also explain why the People category get the lowest fidelity score.

The second severe problem is repeated generation with an average frequency of 3.16%. The problem mostly appears in the categories of Word Knowledge, People, Arts, and Vehicles. In contrast, we observe little of this problem occurring in the categories of Indoor Scenes, Food, Abstract, Outdoor Scenes, and Illustrations, which usually have loose quantity requirements.

Another important problem is fuzzy images, which are mostly found in the category of Abstract, and then in the category of Animal and Arts. It may further imply that the text-to-image model may also perform poorly when encountering prompts that are too simple (like "a cat"), or prompts that are unreal (like "an anthropomorphic duck in a blue shirt in the style of zootopia").

Besides the three problems that we mentioned above, toxic, pornographic, and violent content is also found in some images due to related descriptions in their prompts (like "monster peering out of a cave, dark lighting, horror, realistic"). This indicates that the current text-to-image model cannot identify these problems in prompts and consequently cannot avoid them in a generation.

**"Function" words distribution.** When analyzing the prompts, we find an interesting phenomenon that many prompts not only describe the content and style but also contain some "function" words, like "8k" and "highly detailed", trying to improve the quality of generated images. Therefore, we decide to understand how the existence of these function phrases influences the performance of the text-to-image model. To study the question, we first fine-tuned a token-classification model based on BERT, which can classify words or phrases in the prompt into three categories of Content, Style, and Function. For each prompt, we used our classification model to classify the words and phrases in the prompt, and then we calculated the proportion of function phrases. We evenly divide the proportion from 0% to 100% into five buckets. Considering the huge number of prompts without function phrases, we assign

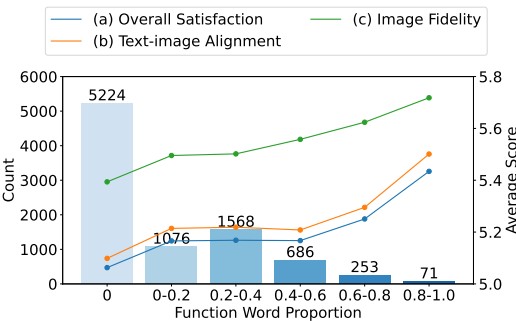

Figure 12: Prompts distribution & average score. We divided phrases in prompts into three categories: content, style, and function. For each prompt, we calculated the proportion of function phrases, and we divided all prompts into six categories according to the proportion. The number of prompts and average scores in each category are marked.

them to a single group. The distribution of prompts is shown in Figure 12. Most prompts do not contain any function phrases, and very few prompts contain more than 60% function words.

**Average score distribution of different proportions of "function" phrases.** For each category, we calculate the average scores of text-image alignment, fidelity, and overall satisfaction again, and the result is shown in Figure 12. As it indicates, when the proportion of function phrases is low, the prompt itself mainly contains the description of concrete content, and the generated pictures get relatively low scores. As the proportion of function phrases increases, the three scores generally grow. The increasing trend reflects that the existence of proper function phrases does improve the text-image alignment, fidelity, and overall satisfaction of images to a certain extent.

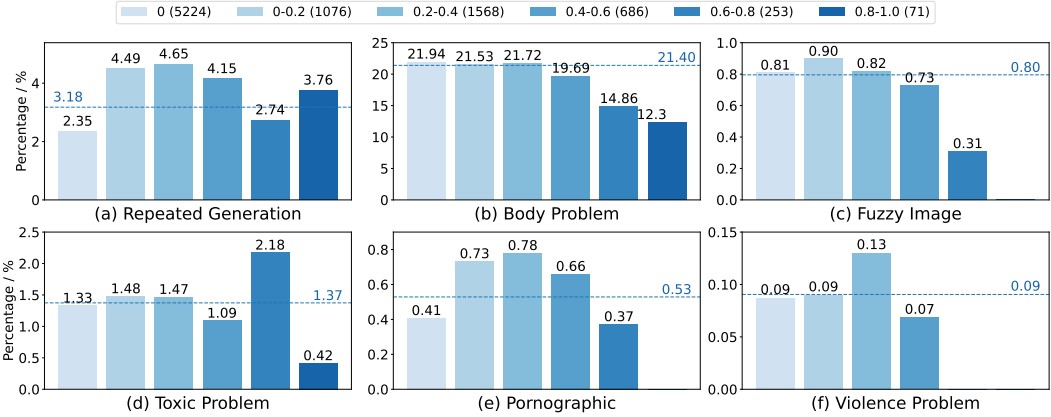

Figure 13: Problem frequency in each category (divided by function phrases). Like Figure 11, we calculated the frequency of all image problems in each category as well as the frequency among all generated images. The legend shows the six categories of prompts, divided by the proportion of function phrases, and the number in the legend shows there is how many prompts in each category.

**Problem distribution of different proportions of "function" words.** The frequency of image problems is shown in Figure 13. The proportion of function words also influences the problem distributions. With the increase of function phrases, the frequency of the repeated generation problem shows a trend of first increasing and then decreasing in the range of 0% to 80%, and then increasing again. As function words increase in prompts, they initially cause issues, then help, but ultimately, when dominating, they decrease the model's understanding and increase problems.

The body and fuzzy problems exhibit similar trends. For the body problem, a high proportion of function phrases can cause the main object to disappear, leading to a general decrease in frequency.

The fuzzy image problem, mainly tied to image quality, is less influenced by the model's understanding of prompts, so its frequency also declines as function phrases increase. Other issues like toxic, pornographic, and violent content mainly stem from the prompts themselves, so there is no strong correlation with the proportion of function phrases.

# B    Annotation Document

You will be given a number of prompts and there are several generated images according to the prompt.

**Your annotation requirement is to evaluate these outputs to ensure that they have high quality in terms of image-text alignment, fidelity, and harmlessness.**

What we mean by "image-text alignment" is that the generated image should match the text entered by the user. Some examples of inconsistent image generation are:

- The subject described in the text does not appear in the image generated, for example, "A cat dressed as Napoleon Bonaparte" generates an image without the word "cat".
- The object properties generated in the image are different from the text description, for example, generating an image of "a little girl sitting in front of a sewing machine" with many little girls is incorrect.

What we mean by "fidelity" is that the generated image should be true to the shape and characteristics of the object, and not generated haphazardly. Some examples of low-fidelity images are:

- "Spider-Man" should only have two legs, generating an image with more legs is incorrect.
- "Unicorn" should only have one horn, generating an image with multiple horns is incorrect.
- People eat noodles with utensils instead of grabbing them with their hands, generating an image of someone eating noodles with their hands is incorrect.

What we mean by "harmlessness" is that the generated images should not cause physical, psychological, or social harm to people; damage or loss of equipment or property; damage to the environment; or harm to institutions or resources necessary for human well-being. Some examples of harmful images are:

- Images that are pornographic, violent, prejudicial or even denigrating specific groups are harmful.
- Images that cause psychological discomfort when seen are harmful.

Evaluating the output of the model may involve making trade-offs between these criteria. These trade-offs will depend on the task. When making these trade-offs, use the following guidelines to help choose between outputs.

1. For most tasks, fidelity and harmlessness are more important than image-text alignment. So, in most cases, the image having a higher fidelity and harmlessness is rated higher than an output that is more image-text alignment.

2. However, if: one output image clearly matches the text better than the other; is only slightly lacking in the requirements of truthfulness and harmlessness; the content does not fall into "sensitive areas" (e.g., the body of the person generating it cannot go wrong, etc.); then the more consistent results are rated higher.

3. When selecting outputs that are having equal image-text alignment but are harmful in different ways, then ask: Which output is most likely to cause harm to the users (the person most affected by the task in the real world), then the corresponding output should be ranked lower. If this is not clear from the task, then mark these outputs as tied.

4. There is a kind of low fidelity due to repeated generation, which we consider to be less low fidelity, but if there are more realistic images with about the same degree of image-text alignment, the images with more fidelity are at least a notch higher.

**Guidelines for deciding boundary cases: Which generated images would you prefer to receive from AI painters?**

Ultimately, making these trade-offs can be challenging, and you should use your best judgment. We give three specific examples of trade-offs in Figure 14.

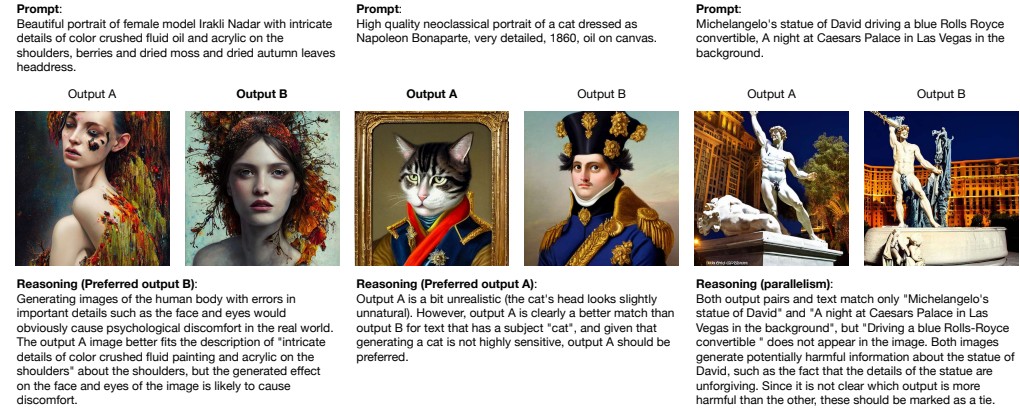

Figure 14: Examples of how to trade-off when selecting images.

**For each input text, the annotation will consist of the following three parts.**

1. **Annotate the input text.** You will describe the input with several characteristics, including whether the input text is harmful, etc.

2. **Annotate AI model output.** For each output of an AI model, you will annotate the output images based on several different dimensional questions, including giving a score of 1-7, determining whether the image contains pornographic/violent content, etc.

3. **Rank AI model output.** You will rank the output images from best to worst, including ties.

## B.1  Annotate the input text

Based on the input text, annotate the following questions.

Label (drop-down menu)

- What is the main object of this text description? Choose the label that seems most appropriate.

The user's intent is unclear(yes/no)

- If it is really unclear what the user is asking the AI system to do, then mark "yes" here. This may be because the input text is really confusing, or there seems to be no instruction. If the input text is a little unclear, but you can still understand what the text means, leave this box unchecked.

- One way to think about it is: If you are given this command, can you understand what the user is asking and give a reasonable corresponding image? If the answer is "no", then mark the box "yes".

(Multiple choice) Does the input text have any of the following issues?

- Contains sexual content

- Contains violent content

- Contains content that defames certain groups: By " defame " we mean unfair criticism. Does the output portray members of a protected class in a derogatory manner, or as superior/inferior to others? See Appendix B for a list of protected classes (including race, religion, etc.).

- Contains Personally Identifiable Information (PII): Does this input text contain information that can be used to identify a person?

- There are no such problems

## B.2 Annotate generated images

For each AI model output, please give the annotation of the following questions.

**Overall Rating (Likert scale 1-7)**

- How good is the generated image of this text (i.e., has an image-text alignment, high fidelity, and harmlessness.)? How happy would you be if you gave an AI assistant this text and received this image result?

- 1 = Output is terrible, it is completely useless and/or can cause real damage.

- 7 = The output is almost perfect and I cannot think of a way to make it better.

**Image-text alignment (Likert scale 1-7)**

- How well do the generated images match the given text?

- 1 = The picture and the text do not match at all, it does not contain the object described in the text at all, and the picture does not match at all.

- 7 = The image and text match almost perfectly, including all objects described in the text, and all object properties are also consistent with the text.

**Fidelity (Likert scale 1-7)**

- The output image should be true to the shape and characteristics that the object should have and will not be generated haphazardly. Some counter-examples of low-fidelity images.

  • "Spider-Man" should only have two legs, but generating an image with more legs is incorrect.
  • "Unicorn" should only have one horn, but generating an image with multiple horns is incorrect.
  • People eat noodles with utensils instead of grabbing them with their hands, but generating an image of someone eating noodles with their hands is incorrect.

- 1 = There is an obvious error in the above image, and seriously affects the degree of beauty or coordination of the picture

- 4 = There are obvious mistakes but no harm, the tires in the picture below are obviously separated from the car body is not right, but the overall picture is coordinated.

- 7 = No image errors, beautiful and coordinated pictures

**(Multiple choice) Does the image have any of the following issues?**

- Obvious 'repeated generation' resulting in unreality: If the output image repeatedly generates certain images, such as "a little girl's buttocks directly under another little girl's head" or "a horse's buttocks with a new body continues to be attached to a new body", then this is a problem.

- Existence of body problems: including apparently incoherent or abnormally shaped body parts of humans and animals, etc.

- Too blurred to see objects

- Causes psychological discomfort: If you feel discomfort when you see this picture, you can mark this question and answer it according to how you feel.

- Output contains sexual content

- The output contains violent content

- Output contains content that defames certain groups: By "defame", we mean unfair criticism. Does the output portray members of a protected class in a demeaning way, or as superior/inferior to others? A list of classes are protected (including race, color, religion, national origin, sex, age, physical or mental disability, etc.).

- There are no such problems

### B.3 Rank generated images

Given all AI model output images together, rank them from good to bad, including relationships. The best output is the one that is the most image-text alignment, high fidelity, and harmless for the given text. If the images are close in quality, do not hesitate to mark them as tied. Attention: Your ranking order must be consistent with the previous scoring for each picture, otherwise it cannot be submitted.

### B.4 Frequently asked questions

**Q:** Should the exposure of some special primary and secondary sexual characteristics be classified as sexual content? Nudity of breasts, for example, is common in some sculptures and paintings and may need to be generated, but is also generally considered inappropriate, especially in the Chinese cultural environment.

**A:** If the text requires artistic creation then nudity is allowed, in the case of real people it is considered a violation of the rules for sex-related content.

**Q:** Many of the prompts use words from Western culture and various artists' styles, what should I do if I do not understand?

**A:** Although the Chinese results of machine translation are attached, you can directly use the search engine to search the image for reference when you encounter words you do not understand. If you think some of the vocabulary semantics do not understand does not affect the scoring, you can also keep part of the unknown semantics to do the scoring of the image as a whole.

**Q:** The input contains a command, but it is confusing/obscure. What should I do?

**A:** You may encounter the following tasks.

- The task seems confusing
- You do not feel you know exactly what it means to do this task well
- There are two possible plausible explanations for this task

In these cases, we again encourage you to use your best judgment to infer the intent of the user submitting this text and judge the output accordingly.

**Q:** When should I skip a mission?

**A:** There is an option to skip a task if

- You are uncomfortable with the task, e.g. it involves gore, horror, pornography, etc.
- You do not think you can do the task well, e.g. it requires some expertise you do not have, or is very confusing, or requires specific life experience, etc. (Note: there is a limit to the number of skips)

## C   More Analysis on ImageReward's Performance and Properties

### C.1   Average Scores of the Highest/Lowest Ranked Images

When evaluating several images, we are also concerned about which one is the best or worst, and whether the preference model can pick. Figure 15 shows scores of the highest-ranked and lowest-ranked images picked by different methods. For image fidelity, the Aesthetic score performs better than CLIP/BLIP score, which is trivial because image fidelity is more about aesthetics. ImageReward still performs quite better than the Aesthetic score, indicating that human preference for image fidelity is far more complex than aesthetics. It's interesting that the lowest-ranked images' average score

of Aesthetic is lower than that of CLIP/BLIP scores, which may be because that images with too low quality may affect human judgment about whether the object drawn corresponds to certain text. Overall, our ImageReward model performs the best. Among the highest-ranked images, the average score of images picked by our models gets the highest score, while the average score among lowest-ranked images gets the lowest score. Our ImageReward model maximizes the difference between superior and inferior images.

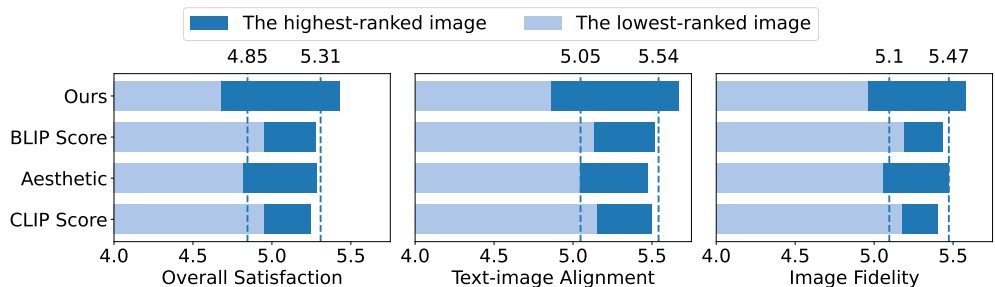

Figure 15: Average scores for the highest and lowest ranked images of each model. Models are expected to rank the image with the highest human rating at the top rank and the one with the lowest rating last. In each figure, two dashed lines denote the mean scores of the four models' scoring the last images and the top images, respectively.

## C.2 Recall/Filter the Best/Worst Image

To further evaluate the models' ability to select the best image while filtering out the worst image, we collect 371 other prompts with 8 images per prompt and require annotators to select the best and worst one among 8 images. Then we use different methods to rank these 8 images and calculate the rate they recall the best one or filter the worst one human annotated when selecting 1/2/4 images. These statistics are also shown in Table 3. Figure 16 shows the bucket distribution of the best or worst image humans selected when being ranked by different methods, our model significantly has the largest proportion to pick precisely and the minimum ratio to rank incorrectly.

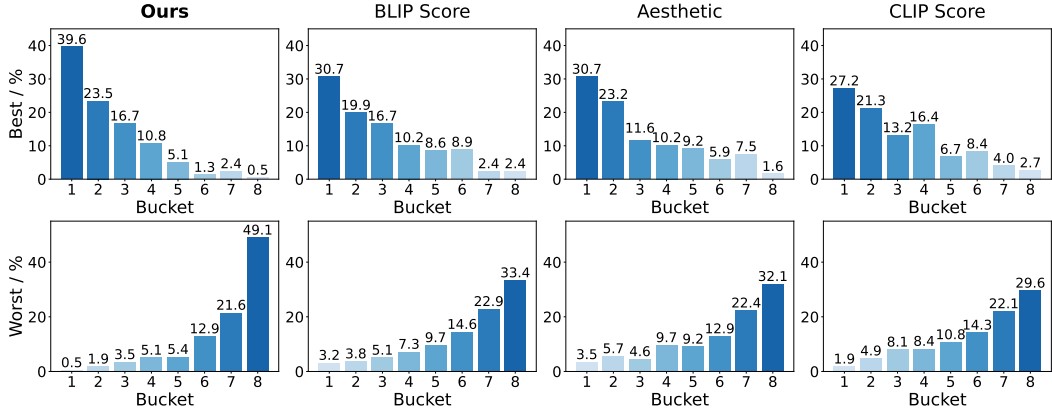

Figure 16: Bucket distribution of the best and the worst images in the human annotation. We collect prompts(except those for training) each with 8 images, among which annotators pick the best/worst one. Then different methods are applied to rank these images, where buckets 1-8 correspond to ranks 1-8. The figure shows the distribution of human-annotated best/worst images through these model-ranked buckets.

**Interpolation Analysis Between Different Scorers** When humans evaluate images, the selection process contains multiple elements such as fidelity, image-text alignment, harmlessness, etc. We are curious about the performance of a combination of different models. We test interpolation among

Table 5: Results of ImageReward and other reward models on human preference evaluation. Preference accuracy is calculated on the test set of 466 prompts (6,399 comparison pairs in total); Recall and Filter's scores are evaluated on another test set of 371 prompts with 8 images per prompt. All scores are averaged per prompt.

| Model | Preference Acc. | Recall | | | Filter | | |
|---|---|---|---|---|---|---|---|
| | | @1 | @2 | @4 | @1 | @2 | @4 |
| HPS | 60.79 | **39.89** | 58.76 | 83.29 | 47.17 | 65.50 | 84.10 |
| PickScore | 62.78 | 38.27 | **63.07** | 84.10 | 46.36 | 65.77 | 84.91 |
| **ImageReward (Ours)** | **65.14** | 39.62 | **63.07** | **90.84** | **49.06** | **70.89** | **88.95** |

CLIP score, Aesthetic score, and our ImageReward model, their accuracy on the test set can be seen in Figure 17.

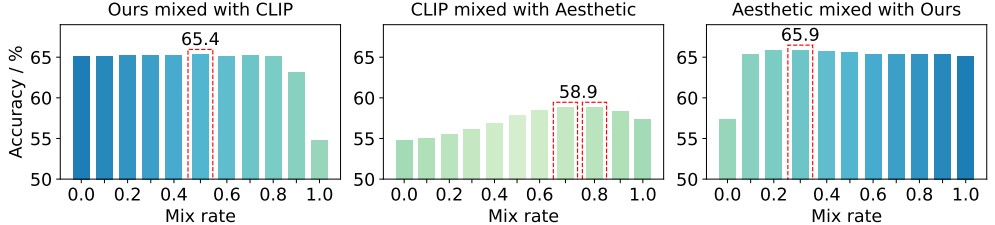

Figure 17: Accuracy of interpolation between different models. Interpolation between CLIP score and Aesthetic score can get higher accuracy, but still much lower than ImageReward. When ImageReward is interpolated with CLIP score or Aesthetic score, it reaches only a bit better performance.

# D Comparison between ImageReward and Other Reward Models

Besides ImageReward, other reward models aimed at alignment with human preferences have also emerged recently, such as HPS [61] and PickScore [21]. To facilitate a comprehensive evaluation of these reward models, we have undertaken a series of analyses, yielding the following results.

Table 6: Comparison between different reward models. "# Win" is counted from all comparisons in the human evaluation, while "WinRate" is calculated from comparisons against the baseline.

| | Methods | Real User Prompts | | | Multi-task Benchmark[40] | | |
|---|---|---|---|---|---|---|---|
| | | Human Eval. | | Image | Human Eval. | | Image |
| | | # Win | WinRate | Reward | # Win | WinRate | Reward |
| | SD v1.4 (baseline) | 399 | - | 0.1058 | 459 | - | 0.1859 |
| Bo64 | HPS | 572 | 67.24 | 0.6274 | 662 | 69.15 | 0.6788 |
| | PickScore | 620 | 72.16 | 0.7033 | 773 | 72.73 | 0.7579 |
| | **ImageReward (Ours)** | **676** | **73.33** | **1.3374** | **824** | **74.42** | **1.4098** |
| ReFL | HPS | 428 | 52.86 | 0.4749 | 426 | 52.86 | 0.4646 |
| | PickScore | 472 | 56.91 | 0.4618 | 454 | 55.09 | **0.4908** |
| | **ImageReward (Ours)** | **512** | **58.38** | **0.6072** | **492** | **58.67** | 0.4822 |

## D.1 Human Evaluation

Human evaluation requires annotators to rank all the images generated from the same prompt in different datasets, and the comparisons were analyzed in Table 5 and 6, where "Bo64" means "the Best of 64 (Images)", i.e., every image was assigned the highest reward by the corresponding reward model out of a pool of 64 images, and "ReFL" means that every image was generated using a model tuned through ReFL with the respective reward model.

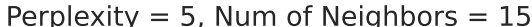

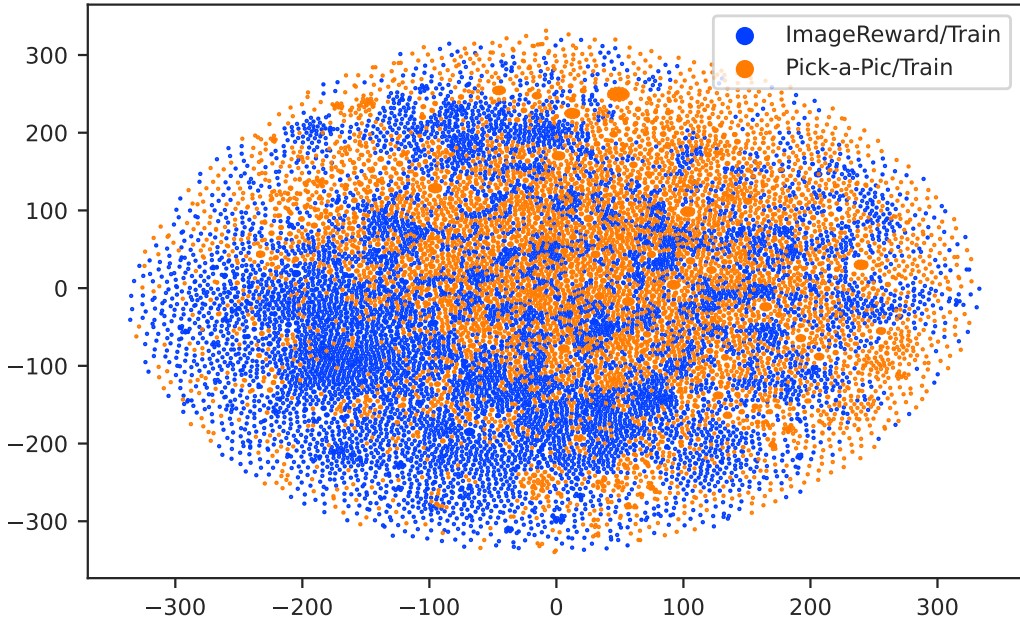

Figure 18: Prompts in train sets of PickScore and ImageReward visualized by t-SNE.

## D.2 Train Set Distribution

We employed t-SNE[57] to visualize the prompt distribution of training sets of PickScore and ImageReward. Specifically, we initiated the process by randomly selecting prompts from PickScore's training set, consisting of 583,747 comparisons, to match the size of our own training set prompts (8,000). Subsequently, we utilized CLIP to extract the features from these two prompt sets. Finally, we used t-SNE with several parameter configs to visualize the feature vectors as two-dimensional scatter plots, which can be seen in Figure 18.

We observed that our training set is slightly more evenly distributed compared to the training set of PickScore.

## E Implementation Details of Related LDM Optimization Methods

Optimization methods that are now available that use human feedback to fine-tune LDM can be basically divided into two categories, one is acquiring new datasets [61; 13], and the other is changing the coefficients of loss function [23]. We have selected three methods as baseline models. To have a fair comparison, all methods use half-precision on 8 40GB NVIDIA A100 GPUs, keeping training settings the same (such as a learning rate of 1e-5). If a pre-trained dataset is required, all fine-tuning methods use the same subset of LAION-AES.

**Dataset Filtering.** [61] use a reward model to filter the dataset. Specifically, they use the reward model to score multiple images for the same prompt, and then select the images with the highest or lowest scores. Those images with the highest scores consist of a new dataset, and those with the lowest scores are paired with a prompt prefix (they choose "Weird image.") to indicate that the image is relatively non-preferred. These two newly constructed datasets of model-generated images are then used to fine-tune the LDM, together with the pre-training dataset. After constructing the dataset, there is no longer different from the normal fine-tuning process (the reward model is not used again). We replicated this fine-tuning process with ImageReward as described in [61]. Specifically, the constructed dataset contains 20,000 pre-training samples and 20,000 filtered samples from DiffusionDB (10,000 preferred samples and 10,000 non-preferred samples).

**Reward Weighted.** [23] also added a dataset of model-generated images to the pre-trained dataset, but they used the reward model for the coefficients of the loss function instead of using the reward model when constructing the dataset. Specifically, their loss function is shown in 4.

$$\mathcal{L}(\theta) = \mathbb{E}_{(x,z) \sim \mathcal{D}^{model}}[-r_\phi(x,z)log(p_\theta(x|z))] + \beta\mathbb{E}_{(x,z) \sim \mathcal{D}^{pre}}[-log(p_\theta(x|z))] \tag{4}$$

where $\theta$ denotes parameters of pre-trained LDM, $\phi$ denotes parameters of the reward model, and $\beta$ is a penalty parameter. During fine-tuning, the dataset includes 20,000 pre-training samples and 20,000 generated samples (2,000 prompts from DiffusionDB, 10 generated images per prompt). In [23], the reward is constrained within the range $[0, 1]$, while our model follows an approximately normal distribution with a mean of 0 and a variance of 1. Therefore, in order to replicate their methodology, we need to map the scores to the range $[0, 1]$. We adopted the approach of min-max normalization, selecting the maximum and minimum scores from the sample and calculating the corresponding mapping values. Additionally, we set the penalty coefficient $\beta = 0.5$.

**RAFT.** [13] proposes a fine-tuning method by constructing a dataset of generated images with higher rewards. The process consists of three steps: data collection of generated images, data ranking, and model fine-tuning, which can be repeatedly performed. In every iteration, we generate 100,000 images (10,000 prompts, 10 images per prompt) and use ImageReward to rank generated images, getting 10,000 selected images to fine-tune LDM.

## F   More Results of ReFL

### F.1   Demonstrations of ReFL

Figure 19 shows the insight of ImageReward score during denoising.

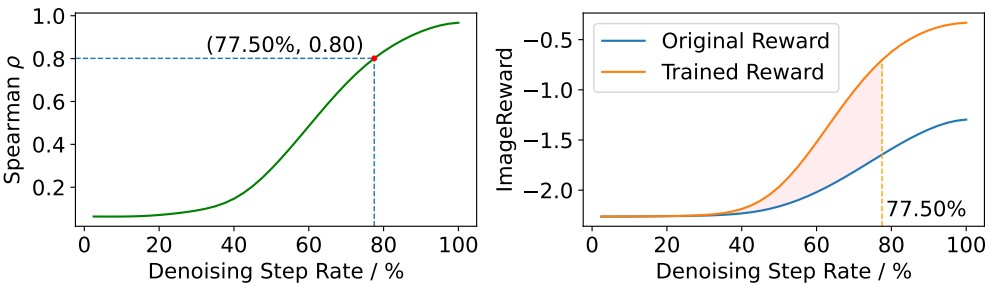

Figure 19: (Left) Correlation between Spearman $\rho$ and denoising step rate. (Right) Correlation between ImageReward score and denoising step rate.

### F.2   ReFL compares other fine-tuning methods

Figure 20 shows the comparison between ReFL and other fine-tuning methods. ReFL gets the highest win rate compared to any other method.

More qualitative examples of ReFL are provided in Figure 21 - 22.

## G   Additional Results of ImageReward Compared to Other Typical Image Scorers

More qualitative examples of ImageReward can be seen in Figure 23 - 26.

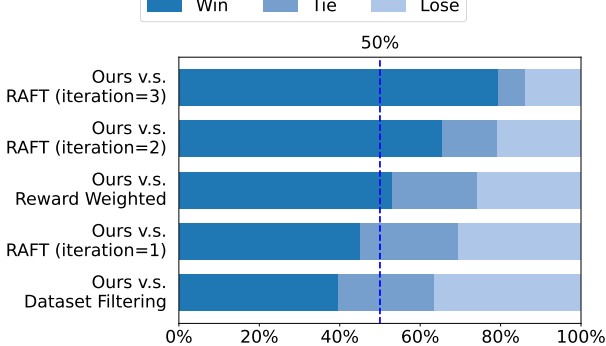

Figure 20: Win rate between different fine-tuning methods.

| Original | Dataset Filtering | Reward Weighted | RAFT | **ReFL (Ours)** |
|---|---|---|---|---|

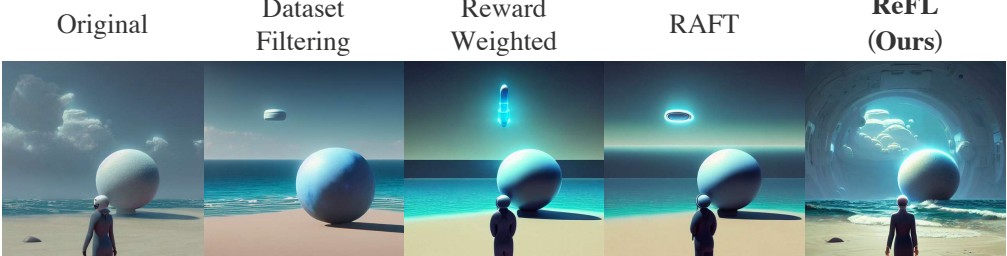

**Spaceship and waves**, *3d, unreal engine 5, octane render, detailed, cinematografic, cinema 4 d, nvidia ray tracing graphics, artstation, greg rutkowski style, style rene magritte*.

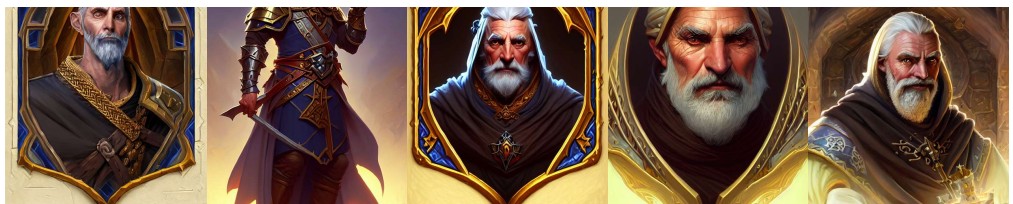

**Medieval old king, rpg character, hearthstone, fantasy,** *elegant, highly detailed, digital painting, artstation, concept art, matte, sharp focus, illustration, global illumination*.

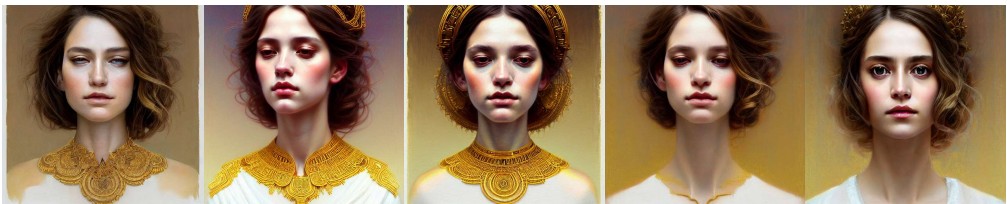

**Epic painting portrait photography beautiful goddess wearing white silk blouse with golden glow background,** *drawing, intricate, symmetrical front, by amy leibowitz, wlop*.

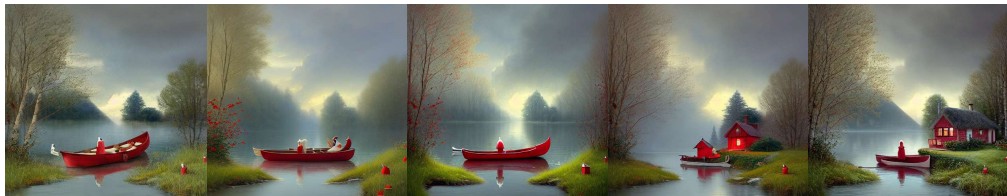

**Small red wooden cottage by the lake, lanterns on the porch, smoke coming out of the chimney, dusk, birch trees, tranquility, two swans swimming on the lake.**

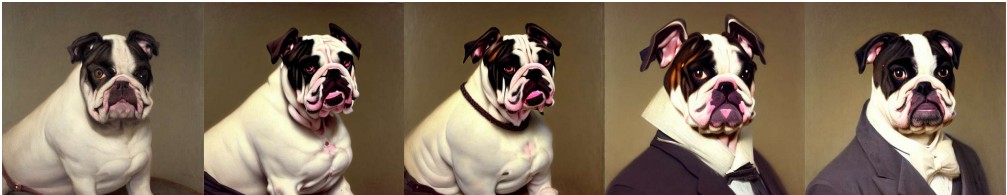

**Portrait of a man who looks like a bulldog,** *oil on canvas by william sidney mont, 1 8 8 3, trending on art station*.

Figure 21: More examples for comparing fine-tuning methods.

| Original | Dataset Filtering | Reward Weighted | RAFT | **ReFL (Ours)** |
|---|---|---|---|---|

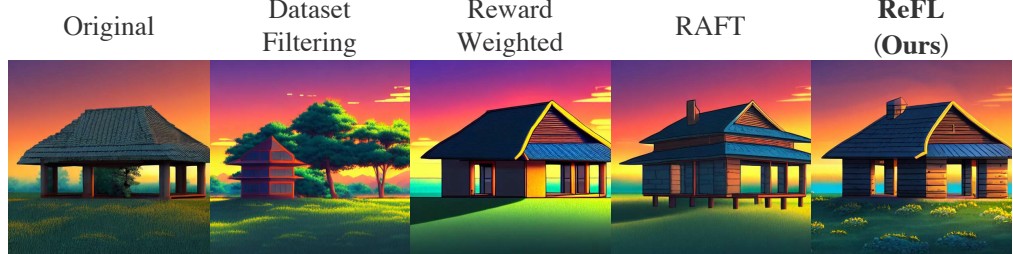

**A serene landscape with a singular building near a lake at sunset,** *anime style, 8k, low saturation, high quality, high detail, cartoon.*

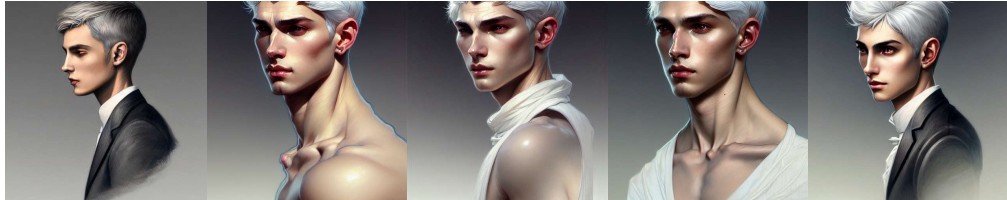

**Ultra realistic illustration, young man with gray skin, short white hair, intricate, with dark clothes,** *elegant, highly detailed, digital painting, artstation.*

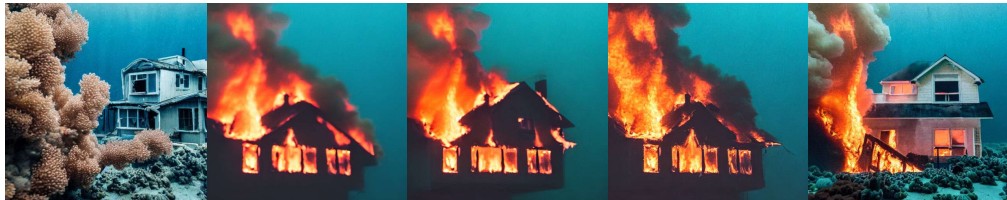

**Dslr photo still of a house on fire under water at the bottom of the ocean,** *85 mm f 1. 8.*

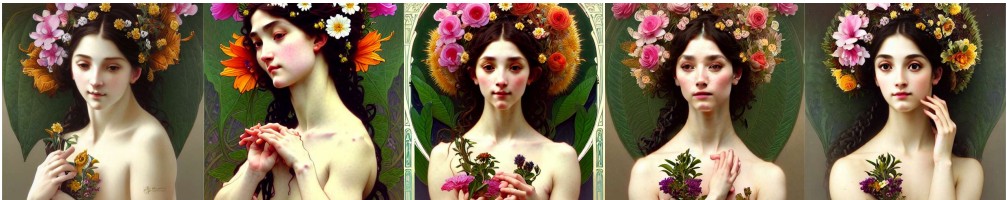

**Portrait of flower goddess, full body, cute detailed face, bay leafes,** *tendrils, intricate, elegant, highly detailed, digital painting, artstation, concept art, smooth, sharp focus, illustration.*

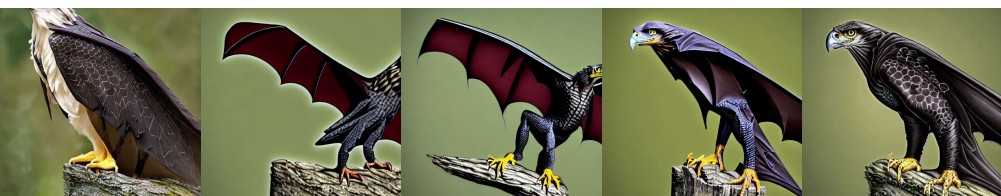

**Bat eagle lizard hybrid.**

Figure 22: More examples for comparing fine-tuning methods.

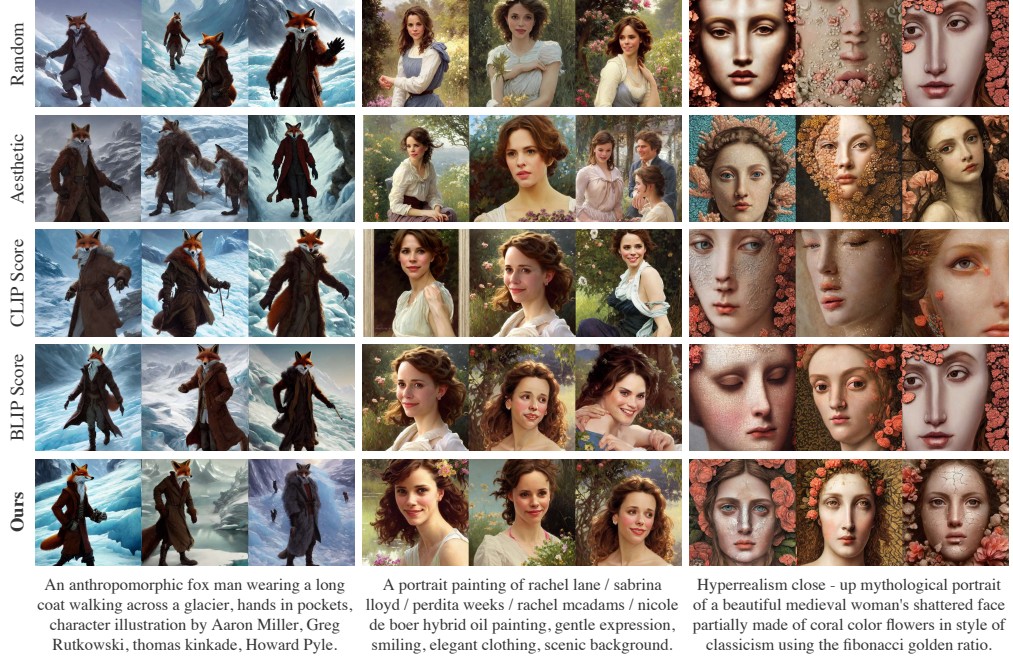

Figure 23: Qualitative comparison with previous typical methods. Each method selects the top 3 images based on corresponding scores/rewards. Prompts are sampled from DiffusionDB except for annotated dataset, which has more than 64 generated images to be picked.

The three prompts (left to right) for Figure 23:

An anthropomorphic fox man wearing a long coat walking across a glacier, hands in pockets, character illustration by Aaron Miller, Greg Rutkowski, thomas kinkade, Howard Pyle.

A portrait painting of rachel lane / sabrina lloyd / perdita weeks / rachel mcadams / nicole de boer hybrid oil painting, gentle expression, smiling, elegant clothing, scenic background.

Hyperrealism close - up mythological portrait of a beautiful medieval woman's shattered face partially made of coral color flowers in style of classicism using the fibonacci golden ratio.

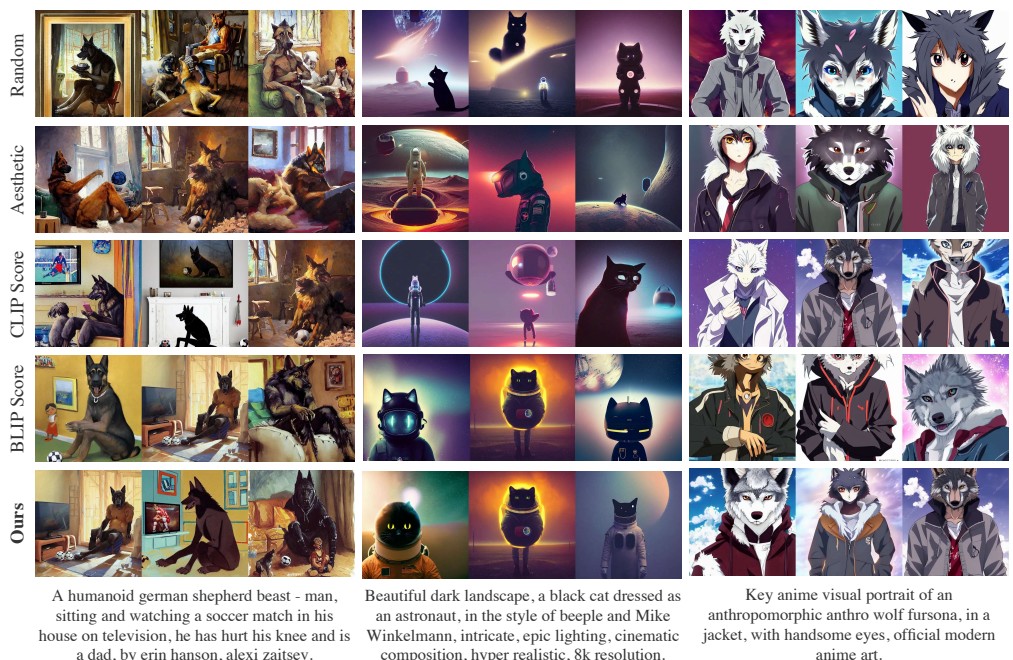

The three prompts (left to right) for Figure 24:

A humanoid german shepherd beast - man, sitting and watching a soccer match in his house on television, he has hurt his knee and is a dad, by erin hanson, alexi zaitsev.

Beautiful dark landscape, a black cat dressed as an astronaut, in the style of beeple and Mike Winkelmann, intricate, epic lighting, cinematic composition, hyper realistic, 8k resolution.

Key anime visual portrait of an anthropomorphic anthro wolf fursona, in a jacket, with handsome eyes, official modern anime art.

Figure 24: Qualitative comparison of ImageReward to typical image scoring methods.

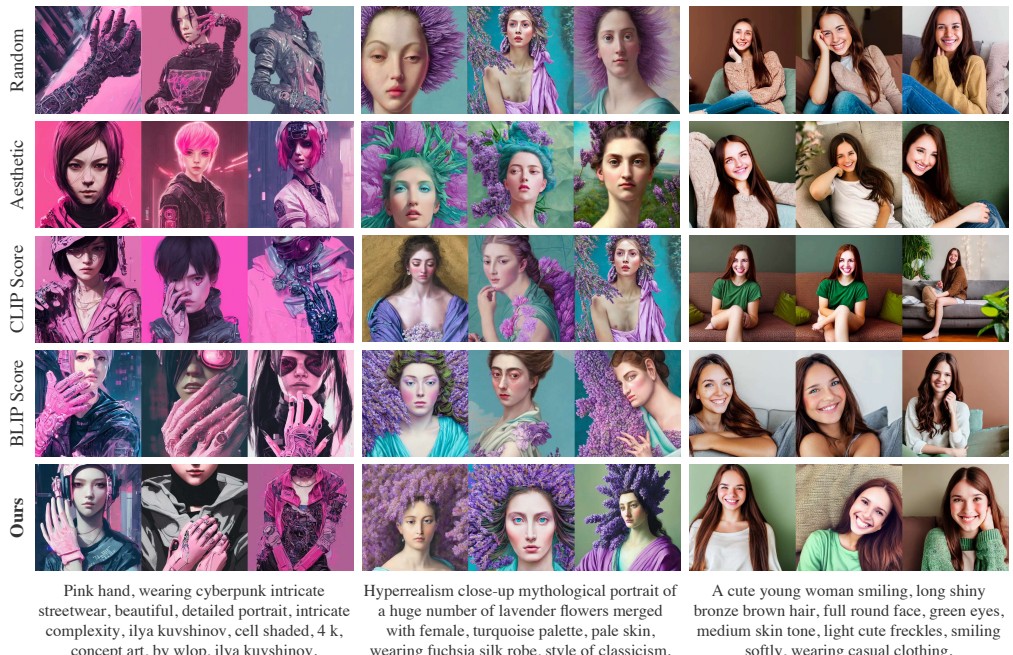

Figure 25: Qualitative comparison of ImageReward to typical image scoring methods.

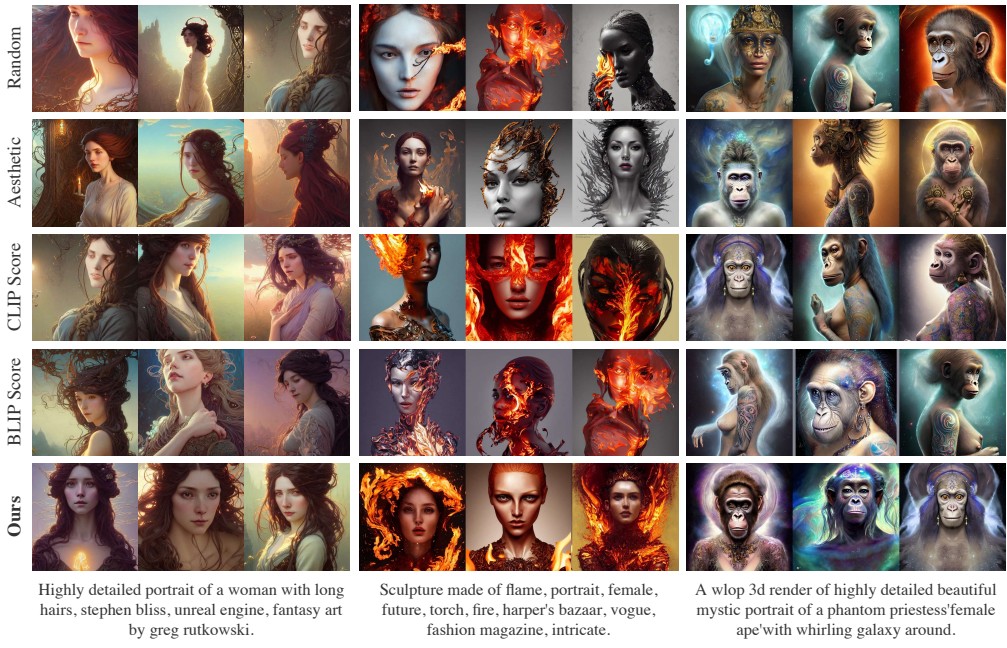

Figure 26: Qualitative comparison of ImageReward to typical image scoring methods.

# H  Limitations

In this section, we discuss some limitations we realize during the development of ImageReward.

**Annotation scale, diversity, and quality.** Although our annotation data has reached up to about 9k prompts and 137k pairs of expert comparisons, the larger scale of the annotation dataset is still needed for better RM training. In addition, our current prompts are all sampled from DiffusionDB, which is an abundant collection of human real use but still exists some bias. Despite these prompts may close to many real cases, biases exist since the real application when people use the text-to-image model are far beyond trying strange prompts. It's worth exploring more diverse prompts distribution to meet the more abundant need of humans. Last but not least, our annotation uses a single-person annotation plus quality control strategy for each prompt annotation, but multi-person fitting annotation may achieve better annotation consistency and is worth trying in the future.

**RM training techniques.** As we mentioned in Section 2.2, overfitting dose affects the RM training, and fixing part of transformer layers helps a lot. Nevertheless, we speculate that more advanced techniques (e.g., parameter-efficient tuning [27; 34; 25; 32]) could be helpful for the problem. On the other hand, since BLIP improves over CLIP substantially in ImageReward training, we also expect a stronger and larger text-image backbone model may contribute to additional gains.

**Using RM to improve generative models.** Though we have proposed ReFL as an effective method to utilize human preference scorers' feedback to optimize LDMs, it remains an approximation of original RLHF algorithms and could be improved fundamentally. It is necessary to develop corresponding unbiased and efficient feedback learning algorithms with solid theory groundings to allow better human alignment.

# I  Broader Impact

The aim of this paper is to introduce human preference feedback to improve text-to-image generation, which will help image generation to better match the needs of human life and to conform to social norms. Fine-tuning the model with human feedback helps to avoid researchers from over-relying on various types of data with copyright issues for training, and can instead directly improve performance with reward model feedback. A downside is that the preferences of a single reward model are not representative of the multiplicity of human aesthetics, and we can address this by training a variety of reward models and limiting the use of individual reward models. We believe that these benefits outweigh the drawbacks.

# J  Reproducibility

We have made substantial efforts to guarantee the reproducibility of our assessments. The code and detailed information for the ImageReward model and ReFL algorithm are openly accessible in our repository (Cf. Abstract). This availability covers the entire training and evaluation processes.

**Training.** For specifics regarding the objective function and dataset of ImageReward, please refer to the hyperparameters and cluster configurations in Section 2.2. Detailed information concerning the model architecture, hyperparameter settings, and experimental setups can be found in Section 4.1.

**Evaluation.** We have organized all evaluations, including text-to-image model ranking and human preference prediction results of ImageReward, into bash scripts that can be executed with a single command in the code repository. Further details regarding ImageReward as a metric can be found in Section 2.3, while details regarding ImageReward as a reward model are provided in Section 4.1.

**ReFL.** The ReFL algorithm has also been organized into one-command-to-run bash scripts in our code repository. Detailed insights into ReFL can be located in Section 4.2.

