# OpenReview forum: "ImageReward: Learning and Evaluating Human Preferences for Text-to-Image Generation"
_NeurIPS.cc/2023/Conference — NeurIPS 2023 poster_

### Official Review · Reviewer_6uei · 2023-07-06

**Soundness:** 3 good
**Presentation:** 2 fair
**Contribution:** 3 good
**Rating:** 6
**Confidence:** 4

**Summary:**

The paper presents a human preference model for text-to-image generation and a method to enhance text-to-image models using this preference model. To achieve this, the authors develop a human preference annotation pipeline and create a dataset consisting of generated images and human ratings. The proposed model is trained to predict human preference rankings, and experimental results indicate that it aligns better with human preferences than existing automatic measures. Additionally, the paper introduces a learning method to fine-tune a diffusion model using the human preference model. The experimental result demonstrates that the text-to-image generation model, when adjusted with the proposed method, is preferred by human annotators.

**Strengths:**

- The paper's clear contribution is the human preference dataset, which features high-quality annotations from a professional data annotation company.
- The experiment suggests that ImageReward outperforms popular measures like FID and CLIP scores in evaluating text-to-image generation.
- By training the text-to-image model using the human preference model, the authors achieve improved performance in both automatic and human evaluations.

**Weaknesses:**

- The paper exceeds the page limit. The authors should carefully follow formatting instructions and revise the manuscript accordingly.
- The description of ImageReward training lacks detail, which may make reproduction difficult.
- Information regarding the human evaluation of experiments in Section 4.2 is missing.
- Certain aspects are unclear. For example, the intention of Figure 7 is unclear for me. What image generation problems are demonstrated in the examples? The numbers in Table 2 are unexplained. What agreement measure is used?

**Questions:**

- Will the entire human preference dataset be made publicly available? The repository looks to provide only test set.
- Can the authors clarify the unclear points mentioned in the Weaknesses section?

**Limitations:**

The limitations section addresses three main issues:
- Annotation scale, diversity, and quality
- The heuristic nature of RM training
- The lack of theoretical background for training the diffusion model with RM feedback
The authors provide meaningful suggestions for future research directions.

---

> ### Author Rebuttal · Authors · 2023-08-10
>
> Thank you for your valuable comments and suggestions.
>
> ### Weakness 1: Formatting issues
>
> We sincerely apologize for the unintentional formatting issues and would appreciate your leniency towards these easily rectifiable issues. In the final version, we will promptly correct these formatting problems by relocating the "Limitations" and "Broader Impact" sections from page 10 to the "Appendix".
>
> ### Weakness 2: Details on ImageReward Training
>
> There might be some misunderstandings about this part. In fact, we have provided detailed information on the necessary aspects to reproduce the training in the following sections of the paper and supplementary code repos:
> 1. In "Section 2.2 RM Training," we provide a comprehensive description of the objective function and how the dataset is utilized.
> 2. In the "Training Setting" paragraph of Section 4.1, we offer detailed information about the model architecture, hyperparameter settings, and experimental setup.
> 3. One-click script to reproduce our experiments: we provide all codes related to reproduce ImageReward training in our code repository (see our abstract), which ensures reproduction with one line of commands. (You can find it at train/src/train.py in the code repository.)
>
> Thanks for your pointing out and  for clarification, we will sum up all the reproduction details in a new “Reproduction” section in the final version.
>
> ### Weakness 3: Details on human evaluation of ReFL (Section 4.2)
>
> Thank you very much for your attention to the details of our experiment! Our human evaluation in Section 4.2 is consistent with Section 2.3 and the form of dataset labeling, which involves humans sorting multiple images under a prompt. We account for the source and number of prompts used in Section 4.2, with different models controlling the same training process as well as parameter settings.
>
> ### Weakness 4: Details on others
>
> > Certain aspects are unclear. For example, the intention of Figure 7 is unclear for me. What image generation problems are demonstrated in the examples? The numbers in Table 2 are unexplained. What agreement measure is used?
>
> 1. Figure 7 is designed to demonstrate that the LDM generated by ReFL can produce images that are more preferred overall, encompassing aspects such as text-image alignment, fidelity, and aesthetics. For example, in the second row of prompts containing "long pointy ears," only the model fine-tuned with ReFL generates correct ears, while other images either lack ears or have inaccurate representations. This illustrates both alignment and fidelity aspects.
>
> 2. "Agreement" refers to the probability that two individuals have consistent judgments on which of the two pairs is superior.
>
> Thanks for your pointing out and we will update these explanations to the final version.
>
> ### Question 1: Dataset Open-sourcing
>
> > Will the entire human preference dataset be made publicly available? The repository looks to provide only test set.
>
> We have already made the whole 137k human preference dataset public, but due to the double-blind policy we cannot provide the public link here.

---

> > ### Comment · Reviewer_6uei · 2023-08-12
> > **Thanks to your response**
> >
> > Thank you for your responses.
> > The responses addressed my concerns.
> > I also appreciate additional dataset contribution.
> > I am increasing my score.

---

### Official Review · Reviewer_5G85 · 2023-07-06

**Soundness:** 4 excellent
**Presentation:** 4 excellent
**Contribution:** 4 excellent
**Rating:** 8
**Confidence:** 5

**Summary:**

This study presents ImageReward, a general-purpose text-to-image human preference reward model. They have collected 137k expert preference dataset, which contain a lot of rating and ranking annotation.  Additionally, the authors propose Reward Feedback Learning (ReFL), a direct tuning algorithm designed to optimize diffusion models. The performance of ImageReward surpasses that of existing models and metrics.

**Strengths:**

1. The paper is very well presented with clear paper writing and good demonstration.
2. The idea is very novel, which explores directly using the human preference as the supervision signals to tune the pretrained text-to-image generation models
3. they also collect a large-scale high-quality human preference dataset, which can inspire many future works.

**Weaknesses:**

BLIP, being outdated, falls short in generating accurate and comprehensive image descriptions. An alternative approach is to employ more recent models like MiniGPT-4 or LLaVa, which have the potential to produce superior reward scores. These advanced vision-language models not only comprehend the objects within the image but also grasp the emotional and artistic aspects. It would be beneficial if the authors could include a comparative analysis involving these models to further strengthen their findings.


**Questions:**

you may also incorporate evaluation metrics and more complex prompts (e.g. with more spatial grounding") to collect the human feedback.

**Limitations:**

it is well discussed in the paper

---

> ### Author Rebuttal · Authors · 2023-08-10
>
> Thank you for your valuable comments, we will explain your concerns point by point.
>
> ### Weaknesses: More advanced backbone for ImageReward
>
> > BLIP, being outdated, falls short in generating accurate and comprehensive image descriptions. An alternative approach is to employ more recent models like MiniGPT-4 or LLaVa, which have the potential to produce superior reward scores. These advanced vision-language models not only comprehend the objects within the image but also grasp the emotional and artistic aspects. It would be beneficial if the authors could include a comparative analysis involving these models to further strengthen their findings.
>
> Thank you for your great suggestion. Indeed, introducing Visual-enhanced Large Language Models (LLMs) to assist with feedback for Text-to-Image (T2I) is a highly valuable avenue for research and exploration. We would like to elaborate a few of our thoughts and discoveries here.
>
> In general, compared to BLIP, the most important aspect of MiniGPT-4 and LLaVa is the use of LLMs possessing powerful capabilities beyond scoring. We believe LLMs can provide evaluations of T2I results from various dimensions *if properly prompted or fine-tuned*, ranging from accuracy in object attributes, spatial relationships, aesthetics, social ethics, human emotions, and more.
>
> We first study how directly prompting those visual LLMs would help scoring human preference. Take LLaVa as an example in our experiment, by inputting images and appropriate prompts into LLaVA-Lightning-MPT-7B, we derived model ratings from its responses and evaluated their consistency with human rankings following practices in LLM rating [1]. In our testing, 39.22% of pairs couldn't be tested due to the model not providing scores, 31.88% received identical scores from LLaVa, and the remaining 28.9% exhibited an accuracy of 53.60% in aligning LLaVa's scores with human preferences. Interestingly, this result was lower than any baseline's performance in our paper, indicating a gap in LLaVa's zero-shot scoring.
>
> Additionally, we also consider how proper fine-tuning with visual LLMs could help. Unfortunately, due to the short response period and potentially complicated tuning strategies required (as is noted in our Section 2.2), we did not finish the fine-tuning experiments on LLaVa. However, we may share some thoughts on potential outcomes if we make it successfully.
>
> In this work, we primarily compared CLIP and BLIP in the paper since many previous works directly used CLIPScore to assess image-text consistency. It's important to note that we only utilized BLIP's encoder and not its decoder. Specifically, we used ViT-L as the image encoder and a 12-layer BERT as the text encoder, which is comparable in model size to CLIP. The most significant difference between BLIP and CLIP is that BLIP is a single-tower model while CLIP is dual-tower. An insight in our work is that a single-tower model may offer superior joint image-text modeling. We also experimentally demonstrated that BLIP outperforms CLIP on the same dataset scale.
>
> Both BLIP and CLIP do not employ LLMs, and their resource requirements (including training costs, inference overhead, GPU memory requirements, etc.) are lower. As a result, it is likely that with properly tuned visual LLMs, we can better create better human preference RMs for the T2I process. However, it is also a problem that these visual LLMs are somehow even larger than the T2I models themselves (e.g., LLaVa’s 7B v.s. Stable Diffusion’s 1B), which would make the training inefficient.
>
>
> ### Question:
>
> > you may also incorporate evaluation metrics and more complex prompts (e.g. with more spatial grounding") to collect the human feedback.
>
> Thank you for your excellent suggestion! Indeed, exploring more complex prompts is a worthwhile direction for future research. We will consider these improvements in the next version of ImageReward. We will consider more complex prompts such as complex spatial relationships, incorporating textual elements within images, complex object/attribute combinations, intricate grammar structures, and combinations of objects that require imaginative thinking beyond the distribution of the training data. Additionally, prompts can be designed to promote diversity and non-discrimination, among other possibilities. Overall, exploring these avenues holds great promise for future research.
>
> References:
> [1] Zheng L, Chiang W L, Sheng Y, et al. Judging LLM-as-a-judge with MT-Bench and Chatbot Arena[J]. arXiv preprint arXiv:2306.05685, 2023.

---

> > ### Comment · Area_Chair_4gNY · 2023-08-19
> >
> > Dear authors,
> >
> > Thank you for your taking the time to respond to the comments.
> >
> > Dear Reviewer 5G85,
> >
> > After reading the authors' response, do you have any additional thoughts?
> >
> > Best,
> >
> > AC

---

### Official Review · Reviewer_JPbK · 2023-07-07

**Soundness:** 3 good
**Presentation:** 3 good
**Contribution:** 3 good
**Rating:** 6
**Confidence:** 5

**Summary:**

The paper explores human preferences and introduces an ImageReward mechanism, which can be employed for evaluating text-to-image generation. The study further enhances the performance of existing text-to-image generation models through a Reward Feedback Learning (ReFL) approach. The experiments primarily focus on the alignment between the proposed ImageReward and human preference/judgment, which is annotated by real individuals.

**Strengths:**

The problem addressed in this paper is vital as it aims to judge the alignment between text-to-image generation models and human preferences during both training and evaluation. The proposed ImageReward model could effectively guide both the training and evaluation processes of text-to-image generation models.

**Weaknesses:**

There are some concerns about experiments including the considered generative models and the results of the correlations with human preferences/judgments. Please refer to more details in the following.

**Questions:**

1. The number of generative models considered in the experiments seems insufficient. With only six text-to-image generation models used for the evaluation of the proposed ImageReward, the ranking results, as demonstrated in Table 1, may be easily consistent. Moreover, even a slight variation in ranking can lead to significantly different correlation values. To enhance the persuasiveness of the experiments, it would be better to include a more diverse range of generative models, which is more convincing.
2. In Table 1, the results show that the Spearman correlation between the zero-shot FID score and human judgment is only 0.09. With this relatively low correlation, does it suggest that there is little relevance between the FID score and human preferences?

**Limitations:**

The authors discuss the limitations and broader impact in the last two sections of the paper and propose potential solutions and outline future directions for further research and development.

---

> ### Author Rebuttal · Authors · 2023-08-10
>
> Thank you for your valuable comments and suggestions!
>
> ### Question 1: On sufficiency of ImageReward experiments
>
> > The number of generative models considered in the experiments seems insufficient. With only six text-to-image generation models used for the evaluation of the proposed ImageReward, the ranking results, as demonstrated in Table 1, may be easily consistent. Moreover, even a slight variation in ranking can lead to significantly different correlation values. To enhance the persuasiveness of the experiments, it would be better to include a more diverse range of generative models, which is more convincing.
>
> Thanks for your suggestion. There might be some misunderstandings about our evaluation of ImageReward. It is worth noticing that in this work, our evaluation of ImageReward is splitted to two parts:
> Evaluation across images (Table 2 & 3 & 5, Figure 5 & 15 & 16): the major evaluation, where ImageReward is sufficiently evaluated and compared to necessary baselines on judging better generations conditioned on each text prompt.
> Evaluation across models (Table 1, Figure 3): the complementary evaluation, where we endeavor to qualitatively show that ImageReward can also serve as an average metric to compare different T2I models’ quality.
> Our emphasis in this work has been on the evaluation of ImageReward across images, which paves the way for the proposal of ReFL. As a result, we have majorly extensively validated the effectiveness of ImageReward through the first type of evaluation rather than the second.
>
> Additionally, we also actually identify some practical difficulties when evaluating across a large number of T2I models as in the second type. As we obtain human evaluations by model ranking, when the number of models grows large, it becomes very challenging for humans to consistently make effective and reasonable comparisons of images from different models. In practice, when the number of images to rank goes beyond 8, we find human rankings become inconsistent and inaccurate for valid evaluation.
>
> ### Question 2: Relation between FID score and human preferences
>
> > In Table 1, the results show that the Spearman correlation between the zero-shot FID score and human judgment is only 0.09. With this relatively low correlation, does it suggest that there is little relevance between the FID score and human preferences?
>
> Yes, we observed that although some models may have higher FID scores compared to others, the images generated by these models better align with human preferences. We would like to conclude that FID scores insufficiently reflect human preferences, which are in fact very crucial for T2I models’ practical applications.

---

> > ### Comment · Area_Chair_4gNY · 2023-08-19
> >
> > Dear authors,
> >
> > Thank you for your taking the time to respond to the comments.
> >
> > Dear Reviewer JPbK,
> >
> > After reading the authors' response, do you have any additional thoughts?
> >
> > Best,
> >
> > AC

---

> > > ### Comment · Reviewer_JPbK · 2023-08-19
> > >
> > > Thanks for the response. It addresses all of my concerns. I will raise the score.

---

### Official Review · Reviewer_7vWQ · 2023-07-08

**Soundness:** 3 good
**Presentation:** 2 fair
**Contribution:** 3 good
**Rating:** 6
**Confidence:** 3

**Summary:**

The paper introduces ImageReward, a new dataset of human preference over generated images given a text prompt. Human preference for images is rated across three dimensionalities: text alignment, image fidelity, and harmlessness. Using the dataset, they train a reward model to score the generated image and text prompt pair. The reward model consists of a small trainable MLP head over BLIP text-image features. The paper further proposes a baseline method of fine-tuning the generative diffusion model using the score model to increase the generated image alignment with human preferences. The fine-tuning loss is a weighted sum of standard diffusion loss and the one from the scoring model (only at the predicted images at lower timesteps of diffusion).

**Strengths:**

The annotated dataset of human preference is one of the first datasets of this kind and scale. This will help in further research in both text-to-image model evaluation and improving the generations with higher human preference.

Both the reward scoring model and fine-tuning method based on the score model are shown to work on par of better than existing baselines.

The paper consists of extensive analysis and details regarding the dataset annotation, scoring model, and its comparison to recent methods.


**Weaknesses:**

1. One limitation of the dataset might be that it becomes less relevant as the generative models improve. Given that the dataset only consists of generated images with their corresponding text prompt and not a plausible ground truth image for the prompt with the best possible human preference score.

2. Only fine-tuning the model on 0-10 timesteps with the scoring model doesn't seem optimal. Specifically, in cases of object omission, the layout has already been decided in the initial stages of diffusion; thus, the reward score guidance at later stages might not be effective. Is there any ablation or analysis regarding what metrics among fidelity and text alignment improve the most?

3. In Eq2, phi is implemented as a ReLU function, as mentioned in line 260. Probably this should be ReLU over the negative of the score function. Because the higher the score, the better. Or is my understanding incorrect?

4. It would be great to expand on the evaluation setup and metrics, which are sometimes not very clear.

    (a) In line 122, are the 100 real user test prompts different than the ImageReward dataset used to train the scoring model? Similarly, 466 and 371 prompts in Table 3.

    (b) How is the "filter" evaluation metric calculated in Table 3? Does this calculate the number of times the model didn't select the worst image in top-k?

    (c) In Table 4, it's unclear how the evaluation numbers are reported. Does it denote the #winrate for each method out of total N samples (N being the sum of the column), or is it a binary comparison of each method vs the baseline? How many generated images per prompt over the 466/77 prompts were used for the evaluation?

    (d) Some of the baseline, e.g., reward weighted fine-tuning method, performs worse than the baseline in Table 4. Is there any analysis regarding that?


**Questions:**

Minor point: some grammatical issue in line 290, 302 and 360

**Limitations:**

yes

---

> ### Author Rebuttal · Authors · 2023-08-10
>
> Thank you for your valuable comments and suggestions!
>
> ### Weakness 1: Usefulness of the ImageReward dataset
>
> Thank you for your suggestion. However, there might be some misunderstandings about the function of the dataset and how we improve T2I models upon them:
> 1. RMs’ model-agnostic generalization: Regarding the relevance of our dataset after T2I models become improved, we want to clarify that we do not directly use annotated data for fine-tuning a T2I model. Instead, we first train Reward Models (RMs, e.g., our ImageReward), on the annotated data. Once ImageReward learns the general human preference, we then use it to generate rewards for fine-tuning the Latent Diffusion Model (LDM). As a result, ImageReward can help any T2I models to improve human preferences as long as they are imperfect. We also extensively experimented and conducted ablation studies in the paper to show that ImageReward learns general human preferences and has good model-agnostic generalization capabilities (Section 4.1, Table 2-3; Appendix C, Figure 15-17; Section 2.3, Table 1).
> 2. Effect of groundtruth image: In response to the mention of ground truth images, during the preparation of the annotated data, we specifically used the clip-retrieval method to collect a set of real images highly corresponding to text prompts in the ImageReward dataset. Each prompt had generated images from the DiffusionDB as well as images retrieved from the LAION-5B dataset using the clip model. However, during the human annotation process, we observed that many of these LAION-5B real images did not consistently align with human preferences, and including this set of real images in the training data did not improve ImageReward's performance.
>
> Nevertheless, we believe that future work can explore more diverse sources and qualities of the annotated images to address the issue. We appreciate your feedback and will consider these points for further improvements in our research.
>
> ### Weakness 2: ReFL’s effect on layout, fidelity and text alignment
>
> Thank you very much for your feedback. We would like to clarify that it is not accurate to say that we only perform fine-tuning on 0-10 timesteps. More precisely, we conduct fine-tuning on the last 25% timesteps of the denoising process. Admittedly, ReFL may have a larger impact on details rather than layouts of generated images, but it does help improve layout. As the multiple denoising steps share the same Unet model, optimizing the Unet parameters in later steps can also impact the denoising in earlier steps, which majorly influence the layout generation.
>
> In addition to Table 4, we conducted a supplementary human evaluation experiment where five annotators provided scores based on alignment, fidelity, and overall quality dimensions. The results revealed that ReFL outperformed the baseline scores by 0.10 in alignment, 0.29 in fidelity, and 0.14 in overall quality. This indicates that ReFL significantly improves fidelity (i.e., details) but also has a positive impact on alignment (i.e., layout).
>
> |                    | alignment | fidelity | overall quality |
> | ------------------ | --------- | -------- | --------------- |
> | SD v1.4 (baseline) | 5.31      | 5.41     | 5.12            |
> | ReFL (Ours)        | 5.41      | 5.70    | 5.26            |
> | Improvement        | +0.10     | +0.29    | +0.14           |
>
> Additionally, it's worth noting that in the practical use of LDM, switching random seeds is a dominant method to produce various layouts. Therefore, a probably more practical way to find a better layout is to switch random seeds and select the best one using ImageReward scoring, rather than using ReFL.
>
> ### Weakness 3: Typo on Eq2
>
> Thanks for pointing it out. On line 260, it is a typo which should be "objective" instead of "loss." If it were a loss, we would need to include a negative sign. In our code implementation, it is correct.
>
> ### Weakness 4: Clarification on evaluation setup and metrics
>
> (a) Yes, the 100 real user test prompts are different from the ImageReward dataset used to train the scoring model. Specifically, these 100 prompts are sampled from the prompts in the DiffusionDB that are outside the training set. In Table 3, there are 466 prompts specially selected as the test set, and 371 prompts are taken from the test set with corresponding image counts greater than or equal to 8, combined with prompts that were annotated by the researchers in the early stages of the study.
>
> (b) It's not about not selecting the worst image in top-k, but rather selecting the worst image in "worst-k" or "lowest score k".
>
> (c) In Table 4, "# Win" is calculated from all comparisons in the human evaluation, while "WinRate" is derived from comparisons against the baseline. All fine-tuned models generate one image per prompt using the same denoising parameters and random seed for evaluation.
>
> (d)
> Note that both RAFT and Reward Weighted do not collect the prompts used by users in real scenarios for fine-tuning, but prompts used in our review are more widely distributed and complex, so the problems with their methods are more clearly exposed.
> RAFT is constrained by the quality of the constructed dataset.  It is important to note that even expert generators have limitations, and when fine-tuning is performed using prompts sampled from real user data, which can be challenging, there may be instances where the expert generator fails to generate high-quality images.
> In the case of the Reward Weighted method, the coefficient used for the rewards is constrained within the [0, 1] range.  This implies that influence of the non-preferred images is not completely eliminated. Similarly, when utilizing real user prompts, it is likely that there will be non-preferred images (even those relatively the best) in the dataset, which can hinders the effectiveness of the Reward Weighted method.
>
>
> We will update the above content to paper for clarification in the final version. Thanks for your suggestion!

---

> > ### Comment · Reviewer_7vWQ · 2023-08-18
> > **Thanks for the response.**
> >
> > Thanks for the detailed response in the rebuttal. It addresses my concerns. I am keeping my score of weak accept.

---

### Official Review · Reviewer_kP49 · 2023-07-09

**Soundness:** 3 good
**Presentation:** 4 excellent
**Contribution:** 4 excellent
**Rating:** 7
**Confidence:** 4

**Summary:**

This paper aims to improve text-to-image (T2I) from human preference feedback. They first collect a human rating dataset to train their human preference model, ImageReward. With the reward model, they further optimize a pre-trained T2I via the proposed Reward Feedback Learning (ReFL). The experimental results indicate that their ImageReward is more robust than the widely-used CLIP-Score, and ReFL-optimized T2I also performs better than baselines.

**Strengths:**

+ This paper is well-written and easy to follow.
+ The collected dataset is valuable for the V+L community as well as the trained ImageReward, which can help various visual generation tasks (not only T2I).
+ The proposed ReFL pipeline can keep improving the T2I model. Both automatic metrics and human evaluation support the superior performance of their framework.
+ They provide lots of qualitative examples and detailed discussion in the supplementary.

**Weaknesses:**

+ The novelty can be an issue since the human preference-trained reward model and feedback learning are already introduced in large language modeling (LLM). It looks like they just apply the same pipeline from LLM to T2I.
+ It is not easy to collect large-scale human preferences for reward model training. Is there a more efficient way to build ImageReward instead of fully relying on human annotations?
+ A detailed analysis of ImageReward should be considered. For example, how many human preference pairs can lead to how well ImageReward and then how well the final optimized T2I model is.

**Questions:**

Please see the Weakness

**Limitations:**

Since their pipeline relies on the collected labels, the human annotation can contain ethnic issues, which they also discuss in Appendix B.

---

> ### Author Rebuttal · Authors · 2023-08-10
>
> Thank you for your valuable comments, we will explain your concerns point by point.
>
> ### Weakness 1: Novelty of ImageReward & ReFL
>
> Despite drawing inspirations from ChatGPT, our work introduces several crucial novelties for T2I scenarios:
> 1. We are pioneering in the introduction of RLHF to the Text-to-Image (T2I) domain, making the incorporation of human preference novel within T2I.
> 2. ImageReward serves as a pioneering reward model that surpasses previously popular models like CLIP and Aesthetic in its superior understanding of human preference. Furthermore, ImageReward proves to be a better T2I evaluation metric in alignment with human preferences than FID and CLIPScore.
> 3. We have released the earliest human preference T2I dataset and corresponding annotating documents, consisting of 137k pairs of expert comparisons.
> 4. Our ReFL algorithm represents the first instance of directly fine-tuning Latent Diffusion Models (LDMs) using human preference gradients, achieving superior results compared to previous algorithms.
>
> It's important to note that we distinguish ourselves from RLHF for Large Language Models (LLMs) in the following significant ways:
> 1. T2I faces distinctive challenges (aligning models with human preference), requiring careful analysis, as discussed in our introduction. The solutions to these challenges diverge from those in LLMs.
> 2. ImageReward operates across both image and text modalities, unlike LLMs that exclusively utilize language models. Our training process involves intricate parameter tuning and strategies, not amenable to casual training.
> 3. Our ReFL algorithm is distinctly founded on analyses of the diffusion model, differing substantially from Transformer-based LLMs. ReFL is also not an RL-based algorithm as those for RLHF on LLMs.
>
> ### Weakness 2: More efficient ways for human preference annotation collecting
>
> This is an excellent suggestion, as we wholeheartedly agree that collecting a large-scale human-annotated dataset can be challenging. To address this data collection challenge, we think there might be two potential approaches that are worth exploring:
> 1. With the advancement of Visual Large Language Models (LLMs) such as MiniGPT-4, LLaVA, VisualGLM, etc., it might be possible to leverage Visual LLMs for data annotation. However, the effectiveness of this approach remains unverified and requires further research and understanding. Additionally, it might involve interactions between different LLMs.
> 2. On the other hand, another promising approach is to deploy the T2I model as a usable product, similar to Midjourney, where human preference data can be naturally collected during users' interactions with the system.
>
> Nevertheless, building expert annotated dataset and training high-quality image RMs at first is crucial for identifying problems in T2I generation and calibrating automatically created preference data, as which could be noisy and inaccurate.
>
> ### Weakness 3: Detailed analysis of ImageReward
>
> > A detailed analysis of ImageReward should be considered. For example, how many human preference pairs can lead to how well ImageReward and then how well the final optimized T2I model is.
>
> Thank you for your suggestion! In fact, we have provided detailed training specifics of ImageReward in section 4.1, along with result analysis and an ablation study. For instance, the impact of the number of preference pairs, as you mentioned, is addressed in line 245-248 (Ablation Study: Training dataset size), and the numerical results are presented in Table 2b.
>
> ### Ethics Concern
>
> Our work is built upon the foundation of academic research in the field of Text-to-Image (T2I) and involves the exploration of human feedback. Throughout this process, we have considered and taken measures to address potential ethical concerns associated with T2I. Our aim has been to reduce and mitigate existing ethical challenges, rather than amplify them. Specifically, we have taken the following steps to contribute to the reduction of ethical concerns in T2I:
> 1. Starting with the data annotation phase, we have prioritized ethical considerations (details can be found in Appendix B Annotation Document):
>    - We required annotators to identify ethical issues present in generated images, including but not limited to toxicity, pornographic content, or violence.
>    - During the scoring and ranking phase, annotators were instructed to assess the harmlessness of images, giving poorer evaluations to images with harmful content.
> 2. In Appendix A, we analyzed various ethical concerns highlighted by annotators in the labeled dataset, conducting both qualitative and quantitative analyses (see Appendix A.4). These insights can serve as a basis for future research efforts. We strictly adhered to human rights and legal considerations during annotation (see Appendix A.2).
> 3. Our ImageReward model, during experimental evaluations, demonstrates improved handling of toxic content and related issues, thereby better avoiding ethical problems present in baseline models.
> 4. The T2I models obtained through our ReFL algorithm are more adept at addressing pre-existing ethical issues compared to models before fine-tuning.
> We have made substantial efforts to address ethical considerations throughout our research and have sought to contribute positively to the field's ethical discourse by actively mitigating potential issues.

---

> > ### Comment · Area_Chair_4gNY · 2023-08-19
> >
> > Dear authors,
> >
> > Thank you for your taking the time to respond to the comments.
> >
> > Dear Reviewer kP49,
> >
> > After reading the authors' response, do you have any additional thoughts?
> >
> > Best,
> >
> > AC

---

### Decision · Program_Chairs · 2023-09-21

**Decision:**

Accept (poster)

**Comment:**

To improve the image-text alignment, this paper proposes a human preference dataset, provides a reward function aligned with human preferences and tuning method called ReFL.


**strengths**

* Large-scale high-quality human preference dataset for text-to-image generation
* Reward model which can be useful for various downstream tasks
* Tuning method based on learned reward function.

**weaknesses/suggestions**

* Training details for reward learning & investigating efficient labeling
* Detailed explanation for human evaluation

I think the paper makes a nice contribution that the community will find valuable. However, I encourage the authors to think carefully about how to reflect the comments or resolve the questions from reviewers in the camera ready version. Also, please take a closer look at ethics reviews and modify the paper by following suggestions.